# The AAA+ ATPase RavA and its binding partner ViaA modulate *E. coli* aminoglycoside sensitivity through interaction with the inner membrane

Jan Felix [1,6,7,11], Ladislav Bumba [1,2,11], Clarissa Liesche[1,11],
Angélique Fraudeau[1,8], Fabrice Rébeillé[3], Jessica Y. El Khoury [4], Karine Huard[1],
Benoit Gallet [1], Christine Moriscot[5], Jean-Philippe Kleman [1], Yoan Duhoo[1],
Matthew Jessop [1,9], Eaazhisai Kandiah[1,10], Frédéric Barras [4],
Juliette Jouhet [3] & Irina Gutsche [1] ✉

Enteric bacteria have to adapt to environmental stresses in the human gastrointestinal tract such as acid and nutrient stress, oxygen limitation and exposure to antibiotics. Membrane lipid composition has recently emerged as a key factor for stress adaptation. The *E. coli ravA-viaA* operon is essential for aminoglycoside bactericidal activity under anaerobiosis but its mechanism of action is unclear. Here we characterise the VWA domain-protein ViaA and its interaction with the AAA+ ATPase RavA, and find that both proteins localise at the inner cell membrane. We demonstrate that RavA and ViaA target specific phospholipids and subsequently identify their lipid-binding sites. We further show that mutations abolishing interaction with lipids restore induced changes in cell membrane morphology and lipid composition. Finally we reveal that these mutations render *E. coli* gentamicin-resistant under fumarate respiration conditions. Our work thus uncovers a *ravA-viaA*-based pathway which is mobilised in response to aminoglycosides under anaerobiosis and engaged in cell membrane regulation.

Although antibiotics are powerful antibacterial weapons, their widespread and frequently uncontrolled use has led to the emergence of numerous multi-resistant bacteria. Aminoglycosides (AGs) are highly efficient antibiotics against Gram-negative pathogens, yet they cause toxic side effects[1]. Understanding what makes bacteria sensitive to AGs and why AG efficiency is notoriously reduced under anaerobic conditions, often encountered by intestinal pathogens, is essential since administration of smaller AG doses reduces their toxicity towards the host. In two paradigmatic enterobacteria, *Escherichia coli* and *Vibrio cholerae*, the expression of the *ravA-viaA* operon is enhanced by anaerobiosis and sensitises the bacteria to AGs, whereas a deletion of *ravA-viaA* enhances their AG resistance and reduces the AG-mediated

toxic stress[2–5]. The extensively studied ATPase RavA[6–8] belongs to the versatile AAA+ superfamily[9,10], whereas the largely uncharacterised ViaA carries a von Willebrand factor A (VWA) domain[11] and stimulates RavA ATPase activity[6,12]. *E. coli* RavA-ViaA is suggested to act as a chaperone for the maturation of two respiratory complexes - Complex I (Nuo)[3] and fumarate reductase (Frd)[12]. Increased AG uptake may therefore result from facilitated Nuo assembly that contributes to proton motive force generation. However, *E. coli* and *V. cholerae* respiratory pathways are very different, and *V. cholerae* lacks the Nuo complex[13]. In addition, the documented RavA-ViaA-dependent inhibition of Frd activity[12] contradicts the hypothesis that *ravA-viaA* enhances respiration. Furthermore, in vitro RavA is sequestered by the acid

A list of author affiliations appears at the end of the paper. ✉e-mail: irina.gutsche@ibs.fr

stress-inducible lysine decarboxylase LdcI in the form of a huge LdcI-RavA cage proposed to function in the acid stress response upon oxygen and nutrient limitation[6,8,14]. Thus, the molecular mechanism of RavA-ViaA action in bacterial physiology in general, and in AGs sensitisation in particular, appears enigmatic. Our recent study has unexpectedly revealed that LdcI forms supramolecular assemblies under the *E. coli* membrane, purportedly at lipid microdomains[15]. Here we biochemically and biophysically characterise the ViaA protein and its interaction with RavA, and show that, when overexpressed inside *E. coli* cells, both proteins localise to the inner membrane, affect membrane morphology and modify cellular lipid homeostasis. We further reveal that both RavA and ViaA interact with specific lipids, identify the lipid-binding sites and demonstrate that their mutations abolish lipid binding in vitro and strongly attenuates the effect on lipid homeostasis in vivo. Moreover, we introduce these mutations into the *E. coli* chromosome and show that they abrogate the effect of the *ravA-viaA* operon on the AG sensitisation. Altogether, this work reveals a *ravA-viaA*-based pathway that is mobilised in the response of *E. coli* to AGs under anaerobic conditions and participates in the regulation of the bacterial cell membrane. Building upon these results, we propose a possible scenario for the *ravA-viaA* function and open up further research perspectives.

## Results

### ViaA is a dimeric, soluble two-domain protein

The protein sequence of ViaA reveals that it has an N-terminal part (N-Terminal domain of ViaA, NTV) with a predominantly α-helical character that does not show sequence similarity to any other known protein (Fig. 1a, residues 1 – 319). The C-terminal part (C-Terminal domain of ViaA, CTV), starting from residue 320, is predicted to be a VWA domain and contains the characteristic MIDAS motif (Fig. 1a). While studies of the potential role of *E. coli* ViaA in the maturation of Nuo and Frd have been initiated[3,12], no protocols for the preparative purification of ViaA are available, its biochemical/biophysical characterization remains scarce, and binding to its potential interaction partners has only been shown indirectly[3,6,12]. Initial attempts to purify a His-tagged ViaA construct using Immobilized Metal Affinity Chromatography (IMAC) and size-exclusion chromatography (SEC) (Methods) resulted in prohibitively low amounts of purified ViaA protein. However, addition of an N-terminal AviTag to the C-terminally His-tagged ViaA (hereafter named AviTag-ViaA-His) stabilised the protein and enabled its high yield purification for characterisation by multi-angle laser light scattering (MALLS) and small-angle X-ray scattering (SAXS) (Fig. 1b–d, Methods). MALLS experiments with purified AviTag-ViaA-His sample showed a monodisperse peak and yielded a molecular weight (MW) of 123 kDa, corresponding to roughly two times the theoretical MW of the AviTag-ViaA-His construct (60.4 kDa). SAXS studies confirmed the dimeric state of AviTag-ViaA-His in solution and resulted in predicted MWs of 130 and 117 kDa using SAXSMoW[16,17] and ScÅtter[18] respectively. Moreover, the distance distribution function calculated from the SAXS data suggests an elongated particle, while the Kratky plot reveals a degree of flexibility, and corresponds to a Kratky plot expected for a multidomain protein with flexible linker (Fig. 1c, d). Although the dimeric character of the purified AviTag-ViaA-His construct was verified by both MALLS and SAXS, we cannot exclude the possibility that non-tagged ViaA may harbor a different

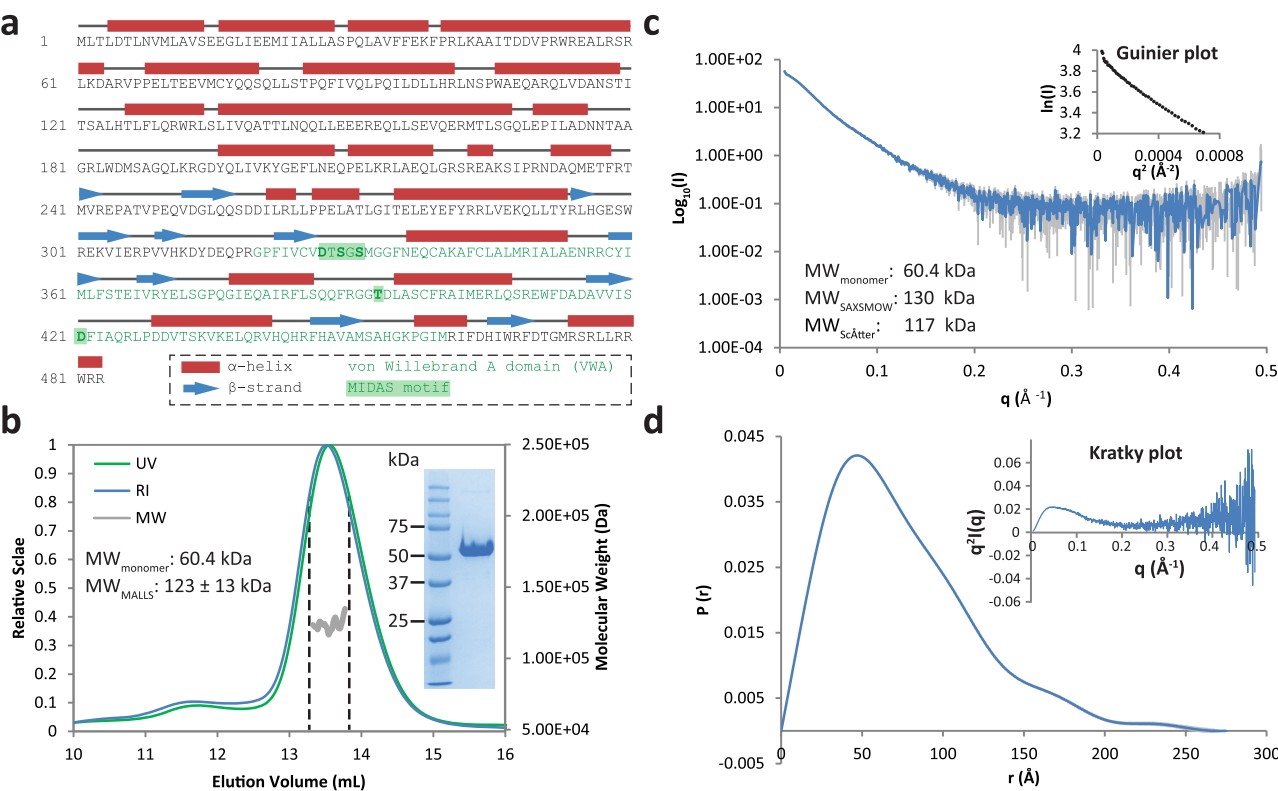

**Fig. 1 | Biochemical and biophysical characterization of ViaA. a** Annotated sequence of *E. coli* ViaA, with predicted α-helices shown as red tubes and predicted β-strands shown as blue arrows. The von Willebrand A (VWA) domain of ViaA is shown in green, with the MIDAS motif highlighted in dark green. **b** Molecular mass determination of AviTag-ViaA-His by SEC-MALLS. The differential refractive index (RI) signal is plotted (left axis, blue curve) along with the UV signal (UV, left axis, green curve) and the determined molecular weight (MW, right axis, grey curve). An SDS-PAGE gel of purified AviTag-ViaA-His is shown as an inset on the right-hand side of the plot. The theoretical monomer MW and the MW as determined by MALLS are annotated on the left-hand side of the plot. The MALLS experiment was performed once. **c** SAXS curve of purified AviTag-ViaA-His. An inset shows a Guinier Plot zooming in on the low-q region of the scattering curve. The theoretical monomer MW as well as the MW determined by SAXSMOW and ScÅtter are shown on the left. **d** Pair-wise distance distribution function *P(r)* and Kratky plot (top right inset) of the AviTag-ViaA-His SAXS data. Source data are provided as a Source Data file.

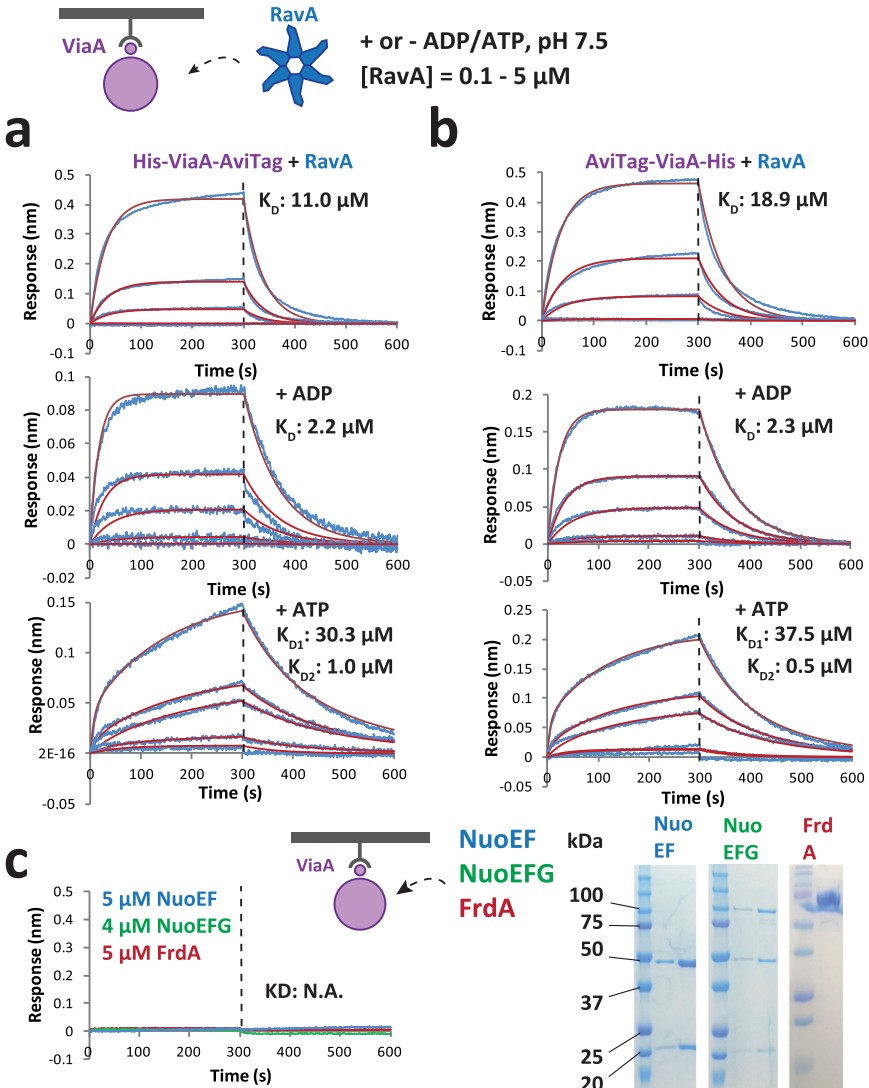

**Fig. 2 | Characterization of the ViaA-RavA interaction by Bio-Layer Interferometry (BLI) binding studies. a** & **b** BLI measurements of His-ViaA-AviTag (**a**) or AviTag-ViaA-His (**b**) coupled on BLI biosensors and RavA, with or without added ADP/ATP. The blue curves correspond to the measured signal while the red curves correspond to the calculated fit using a 1:1 (no nucleotide, ADP) or 2:1 stoichiometry in solution. Furthermore, we note that the elongated heterogeneous ligand binding (ATP) interaction model. **c** BLI measurements of AviTAg-ViaA-His coupled on BLI biosensors and either NuoEF (blue curve), NuoEFG (green curve) or FrdA (red curve). SDS-PAGE gels of purified NuoEF, NuoEFG and FrdA are shown on the right. BLI experiments were performed in triplicate, and one representative experiment is shown. Source data are provided as a Source Data file.

character and apparent flexibility of ViaA might be accentuated by the presence of non-cleaved tags.

## The strength and kinetics of the RavA-ViaA interaction are nucleotide-dependent

While a previous observation that ViaA or NTV slightly enhance the ATPase activity of RavA suggested a physical interaction between the two proteins via the N-terminal part of ViaA[6,12], direct measurements of the RavA-ViaA interaction are unavailable. Pull-down assays using SPA-tagged RavA did not bring down ViaA, and vice versa, pointing to a rather weak or transient interaction[12]. The advantage of AviTag-containing constructs is that they can be biotinylated in vitro and coupled to Streptavidin (SA)-biosensors for binding studies using Bio-Layer Interferometry (BLI). To address potential effects of the protein orientation on the biosensor, either N- or C-terminally tagged constructs (Methods) were immobilised and transferred to wells containing a concentration series of purified RavA in the absence or presence of ADP or ATP (Fig. 2a, b). The affinity of the apo-RavA to immobilised His-ViaA-AviTag and AviTag-ViaA-His could be

described by dissociation constants ($K_D$) of 11.0 μM and 18.9 μM respectively, showing that the location of the AviTag does not have a major influence and validating the weak nature of the RavA-ViaA interaction. ADP binding to RavA increased the strength of the RavA-ViaA interaction to ~2.2 μM, whereas the presence of ATP altered the shape of the BLI curves that could only be reliably fitted using a 2:1 heterogeneous ligand binding model, resulting in $K_{D1}/K_{D2}$ values of 30.3/1.0 μM and 37.5/0.5 μM for His-ViaA-AviTag and AviTag-ViaA-His respectively. The latter may point to the presence of a mixture of RavA conformations after addition of ATP, with different ViaA binding affinities. Taken together, these BLI binding studies revealed that ViaA directly interacts with RavA, and that the strength of this interaction is dependent on the nucleotide-bound state of RavA. ViaA most likely solely interacts with hexameric RavA, in accordance with other characterized AAA+ ATPase and VWA domain-containing protein pairs[19–21]. Since RavA is present as a functional hexamer after incubation with ADP or ATP[8] but has the propensity to partially fall apart into monomers in the absence of ADP/ATP[6], this provides a possible explanation for the notably lower affinity of ViaA for apo-RavA.

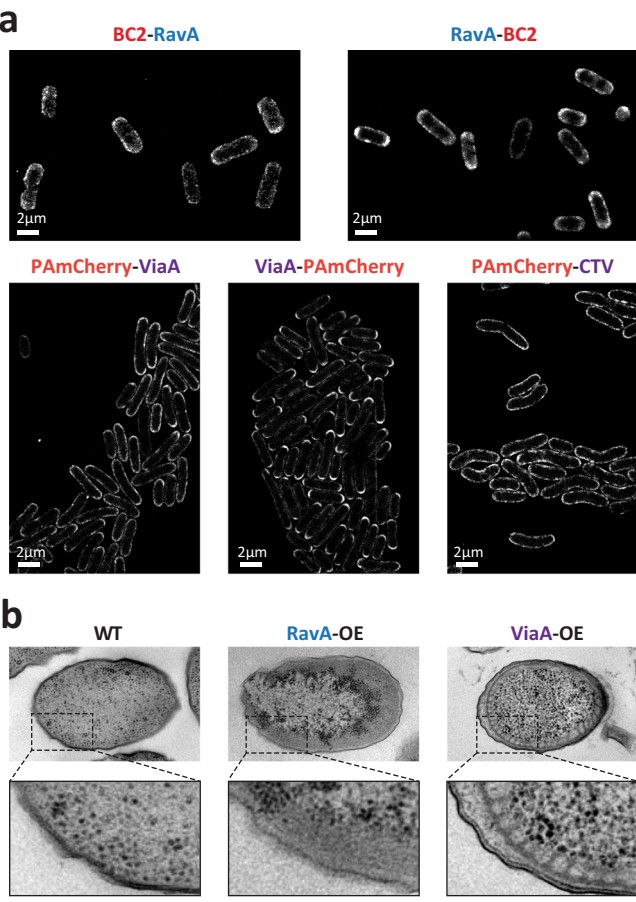

**Fig. 3 | Imaging of overexpressed RavA and ViaA by super-resolution microscopy and cellular TEM. a** Single molecule localisation microscopy imaging of *E. coli* cells overexpressing fluorescently labeled RavA and ViaA by STORM (BC2-RavA, RavA-BC2) and PALM (PAmCherry-ViaA, ViaA-PAmCherry and PAmCherry-CTV). Each experiment was performed in triplicate, and one representative experiment is shown. **b** Imaging of high pressure frozen, freeze-substituted and sectioned wild-type (WT) *E. coli* cells and *E. coli* cells overexpressing His-RavA (RavA-OE) or His-ViaA (ViaA-OE) by TEM. Each experiment was performed in triplicate, and one representative experiment is shown.

Complementary BLI studies using His-NTV-AviTag demonstrated similar RavA binding kinetics as the full-length ViaA despite the monomericity of the NTV construct (Supplementary Fig. 1), confirming that NTV is sufficient for RavA binding[12]. In addition, the absence of the LARA domain of RavA responsible for the LdcI-RavA cage formation[7,14,22] did not affect the binding of RavA to ViaA (Supplementary Fig. 1), indicating that the LARA domain is not involved in the RavA-ViaA interaction.

Previous observation that upon cell fractionation RavA was mostly present in the cytoplasmic fraction whereas the majority of ViaA partitioned into the inner membrane fraction[3], led the authors to the detection of ViaA interaction with specific subunits of Nuo[3] and Frd[12] by pull down assays with SPA-tagged baits. Thus, using BLI, we interrogated the interaction between ViaA and its proposed respiratory complexes' partners: the soluble catalytic NADH dehydrogenase domain of the *E. coli* NADH:Ubiquinone Oxidoreductase I (NuoEFG) and the flavin-containing subunit of the *E. coli* Fumarate Reductase (FrdA)[3,12]. Expression and purification of NuoEFG and FrdA were performed following previously published protocols[23–25] (Methods), and resulted in purified fractions with a red/brown and a yellow color for NuoEFG and FrdA respectively, corresponding to the presence of Riboflavin/Iron-Sulfur clusters for NuoEFG and FAD for FrdA.

However, using concentrations as high as 5 µM of NuoEFG and FrdA did not result in any discernible interaction with either His-ViaA-AviTag or AviTag-ViaA-His (Fig. 2c).

**Both RavA and ViaA show membrane localisation in E. coli cells**
Surprised by the absence of in vitro interactions between ViaA and its purified proposed binding partners, we wondered what might then be the reason for its described partitioning into the inner membrane fraction[3], and therefore sought to localise ViaA and RavA *in cellulo* by single-molecule localisation microscopy imaging. Considering the low amount of natively expressed RavA and ViaA[3], we opted for overexpression (OE) of these proteins with particular tags (collectively called RavA-OE and ViaA-OE for overexpressed RavA and ViaA constructs respectively (See Supplementary Table 1 for the exact nature of different RavA and ViaA constructs used for each experiment presented in this study). The N- or the C-terminus of RavA was tagged with a 12 amino acid BC2 peptide and the resulting RavA-OE was immunolabelled with an anti-BC2 nanobody[26] coupled to the Alexa Fluor 647 fluorescent dye for stochastic optical reconstruction microscopy (STORM) imaging. ViaA-OE constructs were produced by fusion of the PAmCherry fluorescent protein[27] to either the N- or the C-terminus of ViaA, and imaged by photo-activated localisation microscopy (PALM). In addition, PAmCherry was fused to the C-terminus of the NTV and to the N-terminus of the CTV to make NTV-OE and CTV-OE respectively.

Unexpectedly, although RavA is presumed to be cytoplasmic, RavA-OE showed a propensity to localise at the cell periphery rather than being distributed homogeneously through the volume (Fig. 3a). Furthermore, both ViaA-OE constructs, as well as CTV-OE, also exhibited a distinct peripheral localisation pattern around the entire circumference of the cell, with a tendency to accumulate at the polar regions (Fig. 3a). The NTV-OE, in contrast, formed cytosolic bodies (Supplementary Fig. 2). Wide-field images of His-RavA and His-ViaA overexpressing *E. coli* cells labelled with specific anti-RavA and anti-ViaA nanobodies coupled to Alexa Fluor dyes (Methods) further confirmed the inner membrane distribution of RavA and ViaA (Supplementary Fig. 2). Altogether, these experiments indicate that *in cellulo* ViaA is targeted to the bacterial inner membrane despite the absence of ViaA binding to purified NuoEFG and FrdA in vitro, with the CTV being required for this targeting. To determine precise localization of RavA and ViaA in the membrane, an imaging method such as correlative light and electron microscopy (CLEM) would be needed.

**RavA accumulates under the inner membrane whereas ViaA modifies membrane morphology of E. coli cells**
In parallel to imaging the RavA-OE and ViaA-OE bacteria by epifluorescence and super-resolution microscopy, bacteria overexpressing His-RavA, His-ViaA and AviTag-ViaA-His were subjected to high pressure freezing and freeze substitution followed by ultrathin cell sectioning, and their ultrastructure examined by transmission electron microscopy (TEM). A dark layer of matter, supposedly corresponding to the RavA protein, accumulated under the bacterial inner membrane, in line with the fluorescence microscopy observations (Fig. 3b). Most strikingly, arrays of ectopic intracellular membrane tubes were discovered to underlie the inner membrane of the ViaA-OE bacteria (Fig. 3b). Interestingly, overexpression of few membrane proteins with particular topologies has previously been noticed to induce formation of tubular membrane networks of reminiscent hexagonal phase morphologies[28–30]. Such cardiolipin (CL)-enriched neomembranes are proposed to form in order to accommodate the highly overproduced membrane protein, thereby relieving the inner cellular membrane from the associated stress. Thus, on the one side we have biochemically and biophysically characterised a soluble form of ViaA in vitro, and on the other side our morphological cellular EM

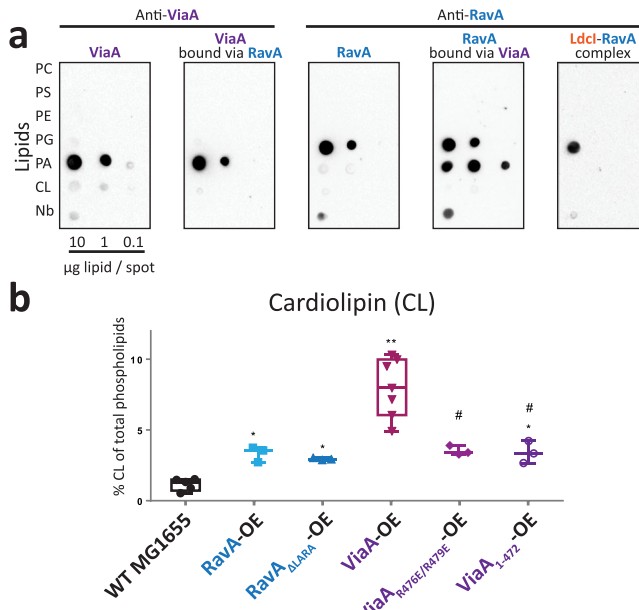

**Fig. 4 | RavA and ViaA binding to specific lipids, and quantification of CL in different RavA and ViaA overexpressing strains. a** Dot-blot assays using purified AviTag-ViaA-His and His-RavA, visualized using anti-ViaA and anti-RavA antibodies and a secondary HRP-antibody conjugate (PC: phosphatidylcholine, PS phosphatidylserine, PE phosphatidylethanolamine, PG phosphatidylglycerol, PA phosphatidic acid, CL cardiolipin). For 'ViaA bound via RavA' or 'RavA bound via ViaA': Lipid binding was first allowed for RavA and subsequently ViaA was added, or vice versa. For 'LdcI-RavA-complex', proteins were mixed before incubation on the membrane (see also Methods). **b** Quantification of cardiolipin (CL) levels by TLC and GC-FID/MS in wild-type (WT) MG1655 *E. coli* cells and MG1655 *E. coli* cells overexpressing His-RavA, His-RavA$_{\Delta LARA}$, AviTag-ViaA-His, AviTag-ViaA$_{R476E/R479E}$-His and AviTag-ViaA$_{1-472}$-His, visualized by a Tukey representation showing all points within a box indicating the median, 25th and 75th percentiles and whiskers down to the minimum and up to the maximum value. Each point represents a biological repeat, with $n = 3$ for all lines except for WT where $n = 5$ and for ViaA-OE (AviTag-ViaA-His) where $n = 7$. The overexpressing (OE) series were compared with the control WT MG1655 series using an unpaired two-sided nonparametric Mann-Whitney test. A significant difference with the control is shown by * for $p$ value < 0.05 ($p$ value = 0.0357 for ViaA$_{1-472}$ and ViaA$_{R476E/R479E}$) or ** for $p$ value < 0.01 ($p$ value = 0.0025). The # indicates a significant difference ($p$ value = 0.0167) with ViaA-OE. Source data are provided as a Source Data file.

observations agree with and corroborate our optical imaging data that show localisation of ViaA and RavA at the inner membrane of the bacterial cell. This raises two immediate questions: (i) are ViaA and RavA capable of directly binding to lipids, and (ii) do they modulate cellular lipid homeostasis, and in particular, does ViaA overexpression increase the cellular amounts of CL? In this respect, it is interesting to note that the conical shape of CL is known to favor its clustering in negative curvature regions such as cell poles and the division septum[31,32], which would agree with the localization of the overexpressed ViaA that we observed by PALM.

## Both RavA and ViaA bind specific anionic phospholipids in vitro

The inner membranes of *E. coli* consist of three major anionic phospholipids: 70-80% phosphatidylethanolamine (PE), 20-25% phosphatidylglycerol (PG) and ~5% cardiolipin (CL), the exact content of the latter being culture condition and growth phase-dependent[33–35]. The universal precursor of these phospholipids is phosphatidic acid (PA), present in extremely low amount in the *E. coli* membrane. To investigate whether ViaA and RavA are capable of directly binding to lipids, we employed dot-blot assays using PE, PG, CL, PA as well as phosphatidylcholine (PC) and phosphatidylserine (PS) (see Methods). While

dot-blot assays are a semiquantitative method, and no direct comparison can be made between intensities of individual dots, they are nonetheless indicative of the relative propensity of a protein to interact with a particular lipid immobilised on a nitrocellulose membrane[36]. Our dot-blots revealed that purified RavA specifically binds PG, whereas ViaA specifically binds PA and to a noticeably lesser extent CL (Fig. 4a, Supplementary Fig. 3). RavA interactions with ViaA and PG are mutually exclusive; likewise, inside the LdcI-RavA complex, RavA is no longer able to interact with PG. In contrast, lipid-bound ViaA is still capable of interacting with RavA (Fig. 4a, Supplementary Fig. 3). Surprisingly, RavA$_{\Delta LARA}$ loses the PG binding capacity (Supplementary Fig. 3), which shows that the LARA domain is implicated not only in the interaction with LdcI but also in the interaction with lipids. In this light, and since the LARA domain is involved in lipid binding but not in ViaA binding (Supplementary Fig. 1), the observation that RavA cannot simultaneously interact with ViaA and PG suggests that binding to PG may induce a conformational change in the RavA molecule, making it inapt for ViaA binding.

The observation that ViaA can simultaneously interact with RavA and with lipids is consistent with ViaA being a two-domain protein, with the NTV binding to RavA and the CTV binding to lipids. Since the 3D structure of ViaA is unknown, we examined its primary sequence and the 3D structure prediction of the CTV provided by Alphafold[37,38] and RoseTTAFold[39] (Fig. 5) for hints to possible positions of a lipid binding site. The C-terminal end of ViaA is highly positively charged, with eight arginines, six of which are situated in the predicted α-helix at the C-terminal extremity (Fig. 5a). Compelled by the amphipatic nature of this helix (Fig. 5b) and its high conservation among enterobacterial ViaA proteins (Fig. 5c), we reasoned that the C-terminus of ViaA might be involved in electrostatic interactions with the negatively charged head groups of PA and CL. In particular, R476 and R479 are predicted to be situated at the same side of the α-helix and seem to form a positive patch (Fig. 5b). To evaluate this hypothesis, we designed the three following mutants: ViaA$_{1-472}$, devoid of the entire C-terminus, ViaA$_{R476A/R479A}$ and ViaA$_{R476E/R479E}$. As shown in Supplementary Fig. 3, mutations of R476 and R479 into alanines notably attenuate interaction with lipids, whereas the removal of the C-terminus and the R-to-E double mutation result in the abrogation of the lipid-binding propensity. Altogether, combined with the membrane distribution and effects on membrane morphology, ViaA unexpectedly joins the list of peripheral membrane proteins tethered to the inner *E. coli* membrane through an amphipathic helix displaying affinity for anionic lipids and in particular CL, which also includes proteins such as the cell division site selection ATPase MinD[40,41] and the cell shape-determining actin homolog MreB[42]. Strikingly, although ViaA is a peripheral membrane protein, its membrane-targeting C-terminal sequence contains all the key features of a high-affinity CL-binding site on bacterial integral membrane proteins, recently identified by molecular dynamics simulations:[43] two or three adjacent basic residues, at least one polar, a neighbouring aromatics and a glycine proposed to confer a higher affinity for the CL head group.

Finally, to further validate the proposed lipid binding sites of RavA and ViaA, we examined the morphology of RavA$_{\Delta LARA}$-OE, ViaA$_{R476E/R479E}$-OE and ViaA$_{1-472}$-OE cells by EM and compared it to the observations presented in Fig. 3b. Remarkably, these mutants formed soluble cytosolic bodies inside the bacterial cells (Supplementary Fig. 4), although the overexpression levels remained very similar to those of the lipid binding-competent counterparts (Supplementary Fig. 5). Thus, the inner membrane localisation and the morphological changes observed with RavA-OE and ViaA-OE are unequivocally related to their propensity to target specific phospholipids.

## ViaA modifies cellular lipid homeostasis in vivo

We then investigated the effect of RavA or ViaA overexpression on the membrane lipid composition (Fig. 4b, Supplementary Fig. 6). To this

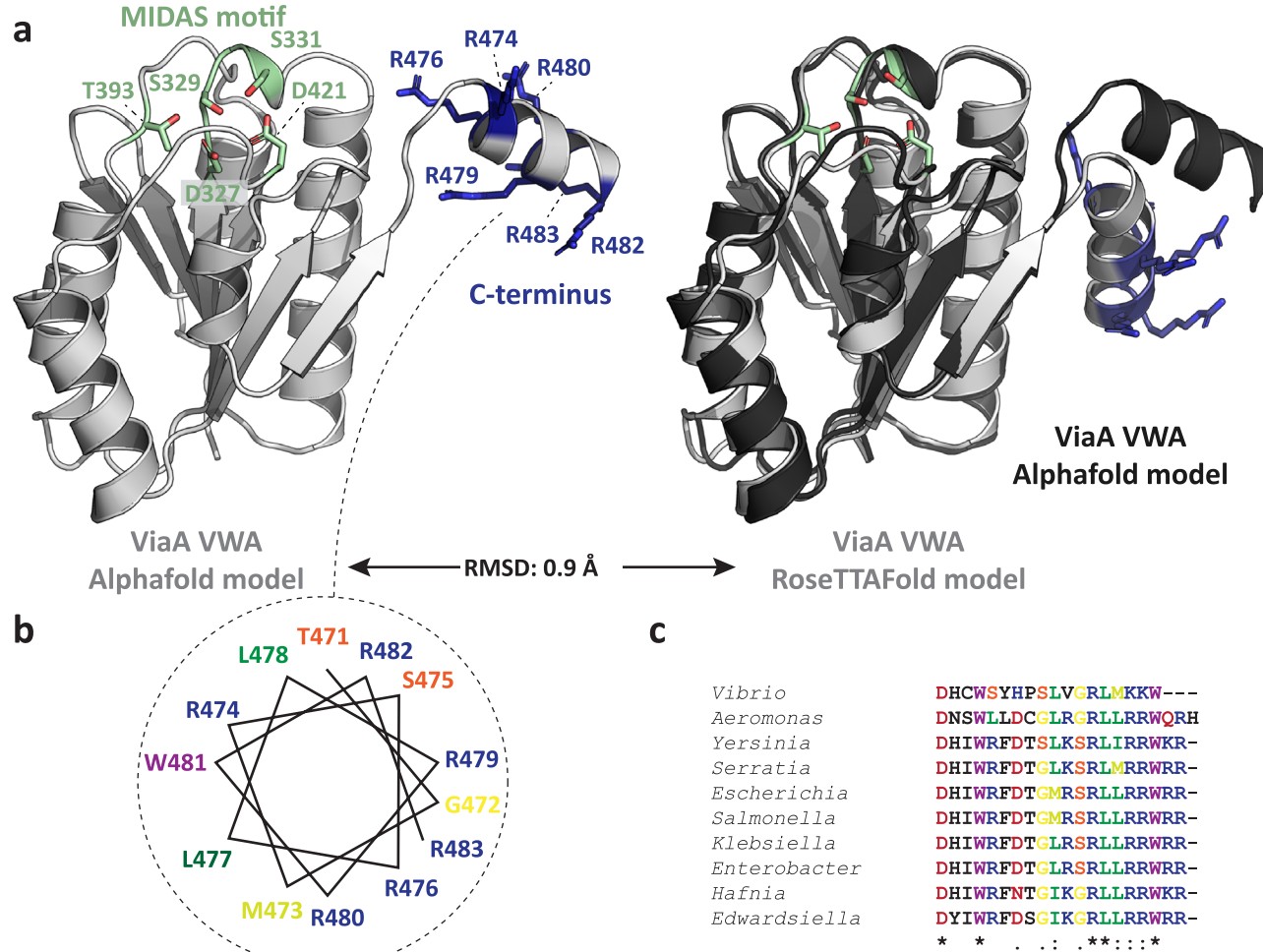

**Fig. 5 | Structure and sequence-based prediction of the ViaA lipid binding site.** **a** Cartoon presentation of Alphafold (left, light grey) and RoseTTAFold (right, light grey) structural predictions for the ViaA VWA domain (residues 320 - 483). The MIDAS motif (green) as well as Arginine residues in the C-terminal helix (dark blue) are annotated and shown as sticks. The Alphafold and RoseTTAFold models are shown superimposed on the right in dark and light grey respectively. The Aphafold prediction of *E. coli* ViaA can be found at https://alphafold.ebi.ac.uk/entry/A0A828TYA7. The majority of the C-terminal VWA domain (155 out of 163 residues, 95.1 %), including most of the C-terminal α-helix, is predicted with a Model Confidence level of 'confident' (90 > pLDDT > 70) to 'very high' (pLDDT > 90). **b** Helical wheel diagram of the C-terminal helix of ViaA (residues 471–483), with annotated and color-coded residues. **c** Multiple sequence alignment of the C-terminal helix of different enterobacterial ViaA orthologues. Residues are color-coded and level of conservation is annotated with '*' (perfect alignment), ':' (strong similarity) and '.' (weak similarity).

end, lipids were extracted from different cellular samples, and analysed by gas chromatography, thin layer chromatography and mass spectrometry (see Methods). As shown in Supplementary Fig. 6, RavA-OE, RavA$_{\Delta LARA}$-OE, ViaA$_{R476E/R479E}$-OE and ViaA$_{1-472}$ -OE bacteria contained slightly but statistically significantly fewer lipids, quantified by their fatty acids, than the wild type (WT) MG1655 cells, whereas the total fatty acid content in ViaA-OE remained roughly similar to the WT. A similar behaviour was observed for the relative proportion of PG among the total lipids, whereas the relative proportion of PE remained unchanged. As anticipated from the electron microscopy observations, the most striking changes were observed with CL, the proportion of which was 6 to 10-fold increased in the ViaA-OE cells, whereas overexpression of the other constructs led to a much milder (but still statistically significant) CL increase in comparison to the WT bacteria (Fig. 4b). One should keep in mind that as observed for some rare membrane proteins[30], an inner membrane stress due to its saturation resulting from the ViaA overexpression may be relieved by induction of the CL-enriched neo-membranes that enclose the excess of ViaA. At physiological concentrations of ViaA however, the CL content of the inner membrane may not be significantly higher than in the absence of

ViaA, and a massive membrane tubulation as observed in Fig. 3b is, at any rate, not to be expected. The low native expression level of ViaA precludes conclusive quantitative assessments of the potential subtle difference in the lipid metabolism between the WT and the MG1655-*ΔravAviaA* strains under anaerobic conditions favoring expression of the *ravA-viaA* operon.

### Lipid binding-deficient chromosomal mutants of RavA and ViaA impair RavA and ViaA function in gentamicin sensitisation

Having recently shown that *ravA-viaA* sensitise *E. coli* to gentamicin (Gm) in fumarate respiratory condition[5], we decided to check if lipid binding-deficient chromosomal MG1655 mutants ViaA$_{1-472}$, ViaA$_{R476E/R479E}$ and RavA$_{\Delta LARA}$ are still capable of Gm sensitisation. To this end, we performed a time-dependent killing experiment, using a Gm concentration equivalent to twice the minimum inhibitory concentration. As shown in Fig. 6, all mutants exhibited increased resistance to Gm compared to the WT strain. This result demonstrates that RavA and ViaA mutants, designed to be unable to bind lipids, prevent Gm toxicity under fumarate respiratory conditions.

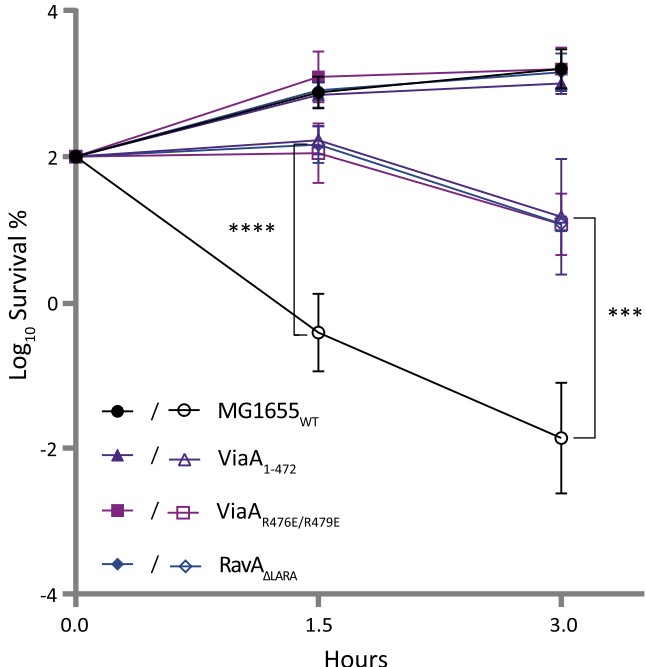

**Fig. 6 | Lipid binding-deficient chromosomal mutants of RavA and ViaA impair RavA and ViaA function in gentamicin sensitisation.** Survival of wild-type MG1655, ViaA$_{1-472}$, ViaA$_{R476E/R479E}$ and RavA$_{\Delta LARA}$ *E. coli* strains after Gm treatment. Cells were grown to an OD$_{600}$ of 0.2 in LB supplemented with 10 mM fumarate, after which 16 μg/mL Gm was added. The survival values after 1.5 and 3 h of treatment are represented. Full and empty symbols are for untreated and Gm-treated bacteria, respectively. Survival, measured by colony-forming units (CFU) per mL, was normalized relative to time zero at which Gm was added (early log phase cells; ~5 × 10$^7$ CFU/mL) and plotted as Log$_{10}$ of % survival. Values are expressed as a mean value of 3 biological replicates and error bars depict standard deviations. A one-way ANOVA test followed by a Dunnett's multiple comparisons test was performed to compare at each time point (1.5 & 3 h) the treated WT to each of the treated mutant strains (*** adjusted *p* value = 0.0002 & **** adjusted *p* value < 0.0001). Source data are provided as a Source Data file.

An attractive hypothesis could be that ViaA, eventually with the help of RavA, would sensitise *E. coli* to Gm by directly binding to the membrane and modifying its permeability. To address this possibility, we performed planar lipid bilayer experiments (Supplementary Fig. 7) but did not detect formation of membrane pores that would facilitate AG transport. Therefore, we concluded that RavA and ViaA target phospholipids without perforating the lipid bilayer such as to directly enable Gm uptake into the bacterial cell.

Finally, we took additional advantage of our time-dependent killing assay to demonstrate that the PAmCherry tag does not affect the properties of ViaA. Indeed, a complementation assay with overexpressed PAmCherry-ViaA and ViaA-PAmCherry (Supplementary Fig. 8) showed that either of the constructs could efficiently restore the impared Gm toxicity for the MG1655-Δ*viaA* mutant strain.

## Discussion

Since AGs are notoriously inefficient under the anaerobic and acidic conditions of the human gastrointestinal tract, insights into the mechanism of the *ravA-viaA* function in *E. coli* sensitisation to AGs should improve our understanding and fight against the global threat of antibiotic resistance. The *E. coli* MoxR AAA+ ATPase RavA has long been known to tightly bind one of the main enterobacterial acid stress response proteins, the acid stress-inducible lysine decarboxylase LdcI, thereby preventing its inhibition by the nutrient stress response alarmone ppGpp[6,7]. However, a characteristic of the MoxR AAA+ ATPases is also the close co-occurence of their genes with genes encoding VWA

domain-containing proteins, with which they are functionally linked[6]. The *ravA-viaA* operon, shown to be controlled by the anaerobic transcriptional regulator Fnr[12], sensitises *E. coli* to AGs in respiratory conditions when fumarate is used as electron acceptor instead of O$_2$[5,44]. Thus, we decided to focus this work on a thorough characterisation of ViaA and the RavA-ViaA interaction, and investigated their molecular functions beyond the LdcI-RavA interaction.

We now provide extensive in vitro and in vivo biochemical, biophysical, optical imaging and morphological evidence that RavA and ViaA do not only interact in a nucleotide-dependent fashion and bind specific phospholipids in vitro, but also localise at the bacterial inner membrane and modulate membrane lipid composition upon overexpression in vivo. In particular, we demonstrate specific RavA-PG and ViaA-PA/CL interaction, and reveal a prominent effect of ViaA overexpression on the CL-enrichment of the *E. coli* inner membrane. A possible explanation of the effect of ViaA-OE on CL can be proposed based on our finding that ViaA interacts with PA (Fig. 4a). Indeed, PA, PG and CL are tightly interconnected: PA, only present in tiny amounts in *E. coli* cells, is a metabolic hub for PG synthesis, whereas two PG molecules are necessary to produce CL[45]. The synthesis of CL in bacteria requires that a first PG receives a phosphatidyl group from a second PG by a transesterification reaction catalysed by cardiolipin synthase (Cls). *E. coli* contains three Cls isoforms[46], with ClsA being the major contributor to CL synthesis during exponential growth. Interestingly, PA was shown to strongly inhibit ClsA activity[47]. Consequently, the association of ViaA-OE with PA at the inner membrane level could trap PA, thereby preventing Cls inhibition and eventually favouring CL synthesis, leading to a modification in the lipid homeostasis. Independently of the concrete mechanism and considering the physiologically low amount of ViaA inside the *E. coli* cell, one may anticipate that by binding to cone-shaped anionic phospholipids such as PA and CL, natively expressed ViaA may locally act on the inner membrane curvature, affect its permeability or influence some other local membrane properties[30].

Importantly, we show that the RavA and ViaA lipid-binding propensity is directly linked to their effect on the AG bactericidal activity under anaerobiosis, because mutations abolishing interaction with lipids preclude Gm toxicity under conditions of fumarate respiration. Our finding that RavA and ViaA do not permeabilise the lipid bilayer, and the independent observations by us and by others that anaerobic RavA-ViaA-mediated AG sensitivity is dependent on the proton motive force (pmf)[3,5,44], suggest that they may positively act on Frd or other respiratory complexes, thereby enhancing the pmf required for efficient AG uptake. Taking into account the absence of a direct interaction with purified FrdA and NuoEFG in a BLI setup as described above, as well as the in vitro lipid binding and the in vivo membrane remodelling activity of overexpressed RavA and ViaA revealed in the present study, we speculate that under physiological respiration conditions, RavA and ViaA might chaperone certain respiratory complexes by acting on lipid microdomains in which these complexes are inserted. This hypothesis aligns with our recent observation of a patchy peripheral distribution of LdcI[15], which we tentatively attributed to an attraction of this proton-consuming acid stress response enzyme to CL-enriched lipid microdomains forming proton sinks[48] that compartmentalise oxidative phosphorylation complexes[49,50]. Thus, this work sets the stage for future investigations of an unprecedented molecular network that links the LdcI-RavA-ViaA triad with bacterial stress adaptation, membrane homeostasis, respiratory complexes and aminoglycoside bactericidal activity.

## Methods

### Bacterial strains and plasmids construction

Constructs used for experiments presented in each of the main and supplementary figures are summarised in Supplementary Table 1. The original plasmids used for creation of new constructs used in this study

are listed in Supplementary Table 2. All novel plasmids, primers, recombinant strains and cloning strategies are summarised in Supplementary Table 3. The *ravA* and *viaA* genes were PCR amplified from *Escherichia coli* K-12 MG1655 genomic DNA using Phusion polymerase (Biolabs). All PCR products were purified by DNA cleanup kit (Qiagen). Gibson assemblies were performed using 0.4 U T5 exonuclease, 2.5 U Phusion polymerase, and 400 U Taq ligase (New England Biolabs) in 1× ISO buffer consisting of 100 mM Tris·HCl pH 7.5, 10 mM MgCl$_2$, 0.8 mM dNTP mix, 10 mM dithiothreitol (DTT), 50 mg polyethylene glycol (PEG)−8000, 1 mM nicotinamide adenine dinucleotide (NAD). A total of 7.5 µL of the Gibson Master Mix was mixed with 2.5 µL DNA, containing ~100 ng of vector. The mix was incubated for 60 min at 50 °C. Transformations were performed in Top10 competent bacteria (One Shot TOP10 chemically competent *E. coli*, Invitrogen) and selected using 100 µg/mL ampicillin or/and 34 µg/mL chloramphenicol (Euromedex). Agarose gel purification and DNA plasmid extractions were performed using a QIAquick Gel extraction kit (QIAGEN). Restriction enzyme digests were performed according to manufacturer recommendations (Biolabs). Bacteria were made electrocompetent by growing cells to an OD$_{600}$ of around 0.7. Cells were then placed on ice and washed three times with ice-cold, sterile water. Electroporation using the MicroPulser (BioRad) was followed by 1 h growth in fresh LB medium.

Mutant strains of K-12 substrain MG1655 (ATCC 47076) were generated by recombineering following the protocols of earlier publications[51,52] and using the recombineering constructs summarised in Supplementary Table 4. Briefly, the pKD46 plasmid was used to express the three proteins Gam, Exo, and Beta, that mediate homologous recombination (HR) and that are originally derived from the bacteriophage lambda. On pKD46, the expression of those genes is controlled by the *araBAD* promoter. pKD46 has an ampicillin resistance gene and a temperature-sensitive origin of replication (ORI), hence cells with pKD46 were grown at 30 °C with ampicillin. Recombineering requires DNA substrates with regions homologous to the target genome. Those substrates were produced by PCR using specifically designed pKD3-plasmids serving as PCR-template. pKD3 encodes the gene *CAT* allowing selection of engineered bacteria using chloramphenicol. The *CAT* cassette is flanked by FRT sites and can be removed using FLP recombinase. FLP is expressed from the pCP20 plasmid. pCP20 has a temperature-sensitive ORI and encodes the ampicillin resistance gene *ampR*, hence cells were grown at 30 °C with ampicillin to keep the plasmid. To induce expression of FLP and to lose the plasmid, cells were grown at 43 °C.

To engineer the here described mutant strains MG1655-*ΔravA/viaA$_{1-472}$viaA(1-472)*::cat, MG1655-*ΔravA/viaA*-$_{R476E/R479E}$::cat and M1655-*ravA$_{ΔLARA}$/ΔviaA*::cat by HR, we targeted the following genomic sequence flanking the *ravA/viaA* operon: (5′-GTATGGCCAGCTGCTGTT CGCGAGAGC GTCCCTTCTCTGCTGTAAGCCATGGTCCATATGAATAT CC-3′) and (5′- CTCGCAATTTACG CAGAACTTTTGACGAAAGGACG CCACTTCATTATGGCTCACCCTCATTTATTAGC-3′). To obtain PCR products for HR that encode mutant *viaA* and lack *ravA*, the corresponding mutant *viaA* genes were cloned into pKD3 (pKD3_ViaA$_{1-472}$), pKD3_ViaA$_{R476E/R479E}$). Likewise, to get PCR products for HR that encode mutant *ravA* and lack *via*A, the corresponding mutant *ravA* gene was cloned into pKD3 (pKD3_RavA$_{ΔLARA}$). For recombineering, the pKD46 plasmid was electroporated into MG1655-*ΔravA/ΔviaA* target cells and grown on ampicillin plates over night at 30 °C. On the second day, single colonies of MG1655-*ΔravA/ΔviaA*:pKD46 were re-streaked on ampicillin plates and grown overnight at 30°C. In the afternoon of the third day, a 20 mL pre-culture of MG1655:pKD46 cells was prepared and grown over night at 30°C. On the fourth day, in order to induce expression of *gam*, *exo* and *beta*, about 3 mL of pre-culture was diluted into 100 mL fresh medium containing ampicillin and 0.2% arabinose (final concentration) and grown to an optical density of about 0.7. Cells were immediately made electrocompetent

and electroporated with the specific PCR product that served as template for homologous recombination. All cells from the electroporation cuvette were plated on chloramphenicol plates and grown overnight at 37 °C. On the fifth day, (all) single clones were re-streaked on fresh chloramphenicol plates for overnight growth at 37 °C. The next day, single colonies were tested by colony PCR and re-streaked on chloramphenicol plates. Growth on test-plates that contain either ampicillin or no antibiotics served as control for expected loss of the pKD46 plasmid and to make glycerol stocks of bacteria respectively. PCR products with the expected size were sent for sequencing (Eurofins). In order to move the recombineered part of the genome into fresh cells, P1 phage transfer was conducted[53].

## ViaA cloning, expression and purification

*E. coli* ViaA (Genebank nr.: AAT48203.1, Uniprot ID: P0ADN0) cloned in the p11-Toronto1 vector and containing an N-terminal hexahistidine (6 x His) tag was obtained from Prof. Dr. Walid Houry. Overlap extension PCR was used to add a C-terminal AviTag (GLNDIFEAQKIEWHE) to the His-ViaA construct (Supplementary Table 3). Additionally, Gibson assembly was used to clone ViaA in the pET22b vector containing a C-terminal His-tag and an added N-terminal AviTag (Supplementary Table 3).

N- and C-terminally AviTagged ViaA constructs were transformed in chemically competent BL21 (DE3) *E. coli* cells. The resulting transformants were grown in a 50 mL preculture supplemented with Ampicillin to an OD of 3.5. Next, 2 L of LB supplemented with Ampicillin was inoculated with 40 mL of the preculture and grown at 37 °C until an OD of 0.8 was reached. Cells were subsequently induced with 0.5 mM IPTG and incubated at 18 °C overnight (ON). The next day, the cells were harvested by centrifuging the cultures at 4,000 g for 45 min.

For purification of expressed AviTag-ViaA-His and His-ViaA-AviTag constructs, pellets (4 for a 2 l culture) were resuspended in 30 mL 25 mM HEPES pH 7.5, 300 mM NaCl, 10 mM MgCl2, 10% Glycerol, 1 µL benzonase, 1 x Complete tablet, and disrupted by three passages through a Homogeniser at 18,5000 psi. Cell debris was removed by centrifugation for 1 h at 48,384 g, and the supernatant was filtered using a 0.2 µL filter and flown over a 5 mL NiNTA IMAC column (GE Healthcare) equilibrated with buffer A (25 mM HEPES pH 7.5, 300 mM NaCl, 10 mM MgCl2, 10% Glycerol, 25 mM Imidazole). AVI-ViaA-HIS was washed with 5 CV of buffer A, and eluted using a gradient of 0–100% buffer B (25 mM HEPES pH 7.5, 300 mM NaCl, 10 mM MgCl2, 10% Glycerol, 300 mM Imidazole). The top fractions after IMAC were pooled and desalted using a 5 mL HiTrap Desalting column (GE Healthcare) equilibrated with 25 mM HEPES pH 7.5, 300 mM NaCl, 10 mM MgCl2, 10% Glycerol. After desalting, the buffer exchanged protein was concentrated and injected on a SD200 10/300 increase column (GE Healthcare) equilibrated with 25 mM HEPES pH 7.5, 300 mM NaCl, 10 mM MgCl2, 10% Glycerol, 1 mM DTT. Purified ViaA fractions, eluted at around 14 mL, were flash-frozen and stored at −80 °C for later use. AviTag-ViaA$_{1-472}$-His, AviTag-ViaA$_{R476A/R479A}$-His and AviTag-ViaA$_{R476E/R479E}$-His were cloned as described in Supplementary Table 3, and purified using the same protocol as for AviTag-ViaA-His.

## NuoEF, NuoEFG and FrdA expression and purification

Expression of soluble NuoEF and NuoEFG was based on Braun et al.[23] and Bungert et al.[24]. Chemically competent *E. coli* BL21 (DE3) cells were transformed with a pET-11a vector containing NuoB-G subunits with NuoF harboring an N-terminal Strep-Tag, kindly provided by Thorsten Friedrich[24]. The resulting transformants were grown for approx. 5 h in a 2 L culture of LB medium supplemented with 100 mg/L ferric ammonium citrate added in aliquots of 20 mg/L per hour, 2 mg/L cysteine, and 20 mg/L riboflavin. Cells were grown at 37 °C and 0.1 mM IPTG was added when the OD reached 0.8. After 3 h, cells were pelleted by

centrifugation at 4,000 g for 10 min. The cell pellet was resuspended in 50 mM MES, 50 mM NaCl, pH 6.6, 10 µg/mL DNase and 1X Complete tablet and disrupted by one passage through a Homogeniser at 18,000 psi. Cell debris was removed by centrifugation for 1 h at 250,000 g. The supernatant was filtered using a 0.2 µL filter and flown over a 5 mL StrepTactin-Sepharose column (GE Healthcare) equilibrated with buffer A (50 mM MES-NaOH, 50 mM NaCl, pH 6.6). The column was washed with buffer A, and the NuoEF and EFG fragments were eluted with buffer A supplemented with 2.5 mM D-desthiobiotin. The eluted NuoEF and NuoEFG mixture appeared reddish/brown, and was further purified using a SD200 10/300 increase column (GE Healthcare) equilibrated with 50 mM MES, 500 mM NaCl, pH 6.6, resulting in two separated peaks containing NuoEFG and NuoEF respectively. Fractions containing NuoEFG and NuoEF, verified by SDS-PAGE, were concentrated separately, flash frozen and stored at −80 °C.

Expression of soluble FrdA was based on the protocol by Léger et al.[25] Chemically competent *E. coli* BL21 (DE3) cells with a knock-out in FrdB were transformed with N-terminal His-tagged FrdA in the pnEA_His6_3C vector, gratefully obtained by Dr. Christophe Romier, Strasbourg. The resulting transformants were used to inoculate 200 mL of LB + 200 µL Ampicillin in a 1 l flask and the culture was grown at 37 °C with vigorous aeration (250 rpm) ON. The pre-culture was used to inoculate 0.5 of LB medium with 500 mL Ampicillin. The resulting culture was grown at 37 °C until the OD reached 0.6. Gene expression was induced with 0.5 mM IPTG. After + − 18 h of expression at 18 °C, cells were pelleted by centrifugation at 4000 g for 45 min. The cell pellet was resuspended in 30 mL Buffer A (50 mM $KH_2PO_4$ pH 7, 500 mM NaCl, 1% glycerol) supplemented with 1x Complete tablet and 0.5 µL benzonase. Cells were opened by sonication, and cell debris was removed by centrifugation for 1 h at 20,000 g. The supernatant was filtered using a 0.2 µL filter and flown over a 5 mL NiNTA IMAC column (GE Healthcare) equilibrated with buffer A (50 mM $KH_2PO_4$ pH 7, 500 mM NaCl, 1% glycerol). Bound FrdA was washed with 5% buffer B (50 mM $KH_2PO_4$ pH 7, 500 mM NaCl, 1% glycerol, 500 mM Imidazole), followed by elution using a gradient from 5–100 % buffer B. Eluted FrdA fractions appeared bright yellow due to the present FAD cofactor. The eluted FrdA fraction was subsequently concentrated and injected onto a SD200 10/300 increase column (GE Healthcare) equilibrated with 20 mM HEPES pH 7.5, 300 mM NaCl, 10% glycerol, 1 mM DTT. Purified FrdA fractions were flash-frozen and stored at −80 °C for later use.

## Multi-angle laser light scattering (MALLS)

Prior to conducting MALLS experiments, flash-frozen purified AviTag-ViaA-His and His-NTV-Avi samples were thawed and centrifuged at 20,000 g for 30 min. SEC-MALLS experiments were conducted at 4 °C on a high-performance liquid chromatography (HPLC) system (Schimadzu, Kyoto, Japan) consisting of a DGU-20 AD degasser, an LC-20 AD pump, a SIL20-AC_HT autosampler, a CBM-20A communication interface, an SPD-M20A UV-Vis detector, a FRC-10A fraction collector, an XL-Therm column oven (WynSep, Sainte Foy d'Aigrefeuille, France) and static light scattering miniDawn Treos, dynamic light scattering DynaPro NANOSTAR and refractive index Optilab rEX detectors (Wyatt, Santa-Barbara, USA).

Purified AviTag-ViaA-His (80 µL at 3 mg/mL) or His-NTV-AviTag (80 µL at 2 mg/mL) were injected on a Superdex 200 increase 10/300 GL column (GE Healthcare), equilibrated at 4 °C with a buffer containing 25 mM HEPES pH 7.5, 300 mM NaCl, 10 mM $MgCl_2$ and 10 % Glycerol, at a flow rate of 0.5 mL/min. Bovine serum albumin (BSA) at 2 mg/mL in phosphate-buffered saline (PBS) buffer was injected as a control. Data was analyzed using the ASTRA software package (Wyatt, Santa-Barbara, USA). The extinction coefficient and refractive index increments for the proteins were calculated from the amino acid sequences using the SEDFIT software.

## Small Angle X-Ray Scattering (SAXS)

SAXS data were collected on the BM-29 BIOSAXS beamline at the ESRF (Grenoble, France)[54], equipped with a Pilatus3 2 M detector operated at a wavelength of 0.9919 Å and using a sample-detector distance 2.867 m, resulting in a scattering momentum transfer range of 0.003 $Å^{-1}$ to 0.494 $Å^{-1}$. Purified AviTag-ViaA-His was measured at a concentration of 3 mg/mL in a buffer containing 25 mM HEPES pH 7.5, 300 mM NaCl, 10 mM $MgCl_2$ and 10 % Glycerol. Measurements were performed at 20 °C, and 10 frames with an individual exposure time of 0.5 s were taken per sample. Initial data integration, averaging and background subtraction were performed using PRIMUS-QT from the ATSAS software package[55].

The forward scattering ($I_0$) and radius of gyration ($R_g$) were determined by PRIMUS-QT using Guinier approximation[56]. The Porod volume estimate ($V_p$) was evaluated using Autoporod[57]. The maximum particle dimension $D_{max}$ and distance distribution function $P(r)$ were evaluated using ScÅtter[18]. The molecular mass of the His-ViaA-AviTag sample was calculated using the online SAXSMoW 2.0 program[17] and the ScÅtter software package.

## Bio-Layer Interferometry (BLI)

For BLI binding studies, biotinylated AviTag-ViaA-His, His-ViaA-AviTag and His-NTV-AviTag samples were expressed and purified as described earlier, but with the addition of 50 µM D-biotin to the LB medium during expression overnight at 18 °C. Binding studies were performed with C-terminally His-tagged RavA and RavAΔLARA proteins purified as described[8]. BLI experiments were performed in HBS kinetics buffer, containing 25 mM HEPES pH 7.5, 300 mM NaCl, 10 mM $MgCl_2$, 10 % Glycerol, 0.1% w/v BSA and 0.02% v/v Tween-20), using an Octet RED96 instrument (FortéBio) operated at 293 K. Streptavidin-coated biosensors (FortéBio) were functionalised with biotinylated AviTag-ViaA-His, His-ViaA-AviTag or His-NTV-AviTag to a maximum signal of 1 nm, subsequently quenched with 10 µg mL$^{-1}$ biocytin, and transferred to wells containing 5 different concentrations of ligand. When RavA or RavAΔLARA was used as a ligand, 1 mM ADP was added to the HBS kinetics buffer and RavA or RavAΔLARA samples were incubated with 1 mM ADP for 10 min prior to conducting measurements. Buffer subtraction was performed using a functionalized biosensor measuring running buffer. To check for nonspecific binding during the experiments, non-functionalized biosensors were used to measure the signal from the highest ligand concentration as well as running buffer. All data were fitted with the FortéBio Data Analysis 9.0 software using a 1:1 interaction model. All binding experiments were performed in triplicate (technical triplicates), and calculated $K_D$, $k_d$ and $k_a$ values represent the averages of these triplicate experiments.

## Nanobody production and labelling

The anti-RavA-Nb and anti-ViaA-Nb were obtained from the nanobody generation platform of the AFMB laboratory (Marseille, France) as described[15]. The anti-BC2-Nb[26] was kindly provided by Ulrich Rothbauer. Nanobodies were labelled as described[15].

## Overexpression of recombinant RavA and ViaA constructs for fluorescence imaging, cellular EM observations, and lipid extraction and analysis

MG1655 cells were transformed with a low-copy auxiliary pT7pol26 (Kan$^R$) plasmid that codes for T7 RNA polymerase under the control of a lac promotor[58]. The resulting MG1655/pT7pol26 (Kan$^R$) strain was then transformed with a plasmid (Amp$^R$) carrying a gene coding for a desired RavA or ViaA construct under the control of the T7 promotor. The cells were cultured at 37 °C until the OD$_{600}$ reached 0.6, and then RavA-OE or ViaA-OE was induced with 40 µM IPTG for 12 h, at 18 °C, under aerobic conditions. For STORM imaging of RavA-OE shown in Fig. 3A, His-BC2-RavA and RavA-BC2-His constructs were used. PALM imaging of ViaA-OE, NTV-OE and CTV-OE shown in Fig. 3A and

Supplementary Figure 2A was performed with His-PAmCherry-ViaA, His-ViaA-PamCherry, NTV-PAmCherry-His and His-PAmCherry-CTV. For wide-field images of RavA-OE and ViaA-OE, His-RavA, His-ViaA and AviTag-ViaA-His constructs were used (Supplementary Figure 2B), and in parallel, an aliquot of each of the cultures was used for cellular EM observations (Fig. 3B). Experiments dedicated to lipid extraction and quantification as well as the assessment of the lipid binding sites (Fig. 4B, Supplementary Figure 4, Supplementary Figure 5 and Supplementary Figure 6) were all performed with the same RavA-OE and ViaA-OE constructs, namely His-RavA, His-RavA$_{\Delta LARA}$; AviTag-ViaA-His (identical results obtained with His-ViaA), AviTag-ViaA$_{1-472}$-His or AviTag-ViaA$_{R476E/R479E}$-His. See Supplementary Table 3 for summary.

### Fluorescence imaging

For immunofluorescence staining cells were harvested, fixed and permeabilised as described[15]. For epifluorescence imaging of fluorescent protein fusion constructs the procedure was the same except that the fixation was performed with 1% formaldehyde solution in PBS, and that no permeabilisation was necessary.

### Wide field imaging

For each sample, 2 µL of cells in suspension were mounted between a glass slide and a 1.5 h glass coverslip, and observed using an inverted IX81 microscope, with the UPLFLN 100× oil immersion objective from Olympus (numerical aperture 1.3), using a fibered Xcite™ Metal-Halide excitation lamp in conjunction with the appropriate excitation filters, dichroic mirrors, and emission filters specific for DAPI/Hoechst, AF488, mCherry or AF647 (4X4MB set, Semrock). Acquisitions were performed with Volocity software (Quorum Technologies) using a sCMOS 2048 × 2048 camera (Hamamatsu ORCA Flash 4, 16 bits/pixel) achieving a final magnification of 64 nm per pixel.

### STORM and PALM imaging

Super-resolution single molecule localization microscopy was performed using STORM (after nanobody labeling) and PALM (PAmCherry fusion proteins) approaches. For STORM imaging, cells labeled as mentioned above, were transferred to a glucose buffer containing 50 mM NaCl, 150 mM Tris (pH 8.0), 10% glucose, 100 mM MEA (mercaptoethylamine) and 1x Glox solution from a 10x stock containing 1 µM catalase and 2.3 µM glucoseoxidase. For PALM, cells expressing the ViaA fused constructs with PAmCherry, were resuspended in PBS. In each case, 2 µL of the cells in suspension were mounted as specified[15]. Mounted samples were imaged on an IX81 microscope (Olympus) by focusing the excitation lasers to the back focal plane of an oil immersion UAPON100X (N.A. 1.49) objective. Acquisitions were performed using an Evolve 512 camera with a gain set to 200 (electron-multiplying charge-coupled device EMCCD, 16 bits/pixel, Photometrics) using Metamorph (Molecular Devices). Excitation lasers power was controlled by an Acousto-Optical Tunable Filter (OATF, Quanta Tech). SMLM datasets of about 30,000 frames were generated using 3 kW/cm$^2$ of a 643 nm laser in conjunction with up to 1 W/cm$^2$ of a 405 nm laser at a framerate of 20 frames per seconds (STORM) or approximately 1 kW/cm$^2$ of a 561 nm laser after photoactivation of PAmCherry using a few watts/cm$^2$ of a 405 nm laser (PALM). Data processing was performed using Thunderstorm plugin in ImageJ[59,60].

### Cellular EM observations

Cell cultures were centrifuged at 4000 g for 5 min. A pellet volume of 1.4 µL was dispensed on the 200 µm side of a type A 3 mm gold platelet (Leica Microsystems), covered with the flat side of a type B 3 mm aluminum platelet (Leica Microsystems), and was vitrified by high-pressure freezing using an HPM100 system (Leica Microsystems). Next, the samples were freeze substituted at −90 °C for 80 h in acetone supplemented with 1% OsO$_4$ and warmed up slowly (1 °C h$^{-1}$) to −60 °C in an automated freeze substitution device (AFS2; Leica

Microsystems). After 8 to 12 h, the temperature was raised (1 °C h$^{-1}$) to −30 °C, and the samples were kept at this temperature for another 8 to 12 h before a step for 1 h at 0 °C, cooled down to −30 °C and then rinsed four times in pure acetone. The samples were then infiltrated with gradually increasing concentrations of Epoxy Resin (Epoxy Embedding Medium kit, Merck) in acetone (1:2, 1:1, 2:1 [vol/vol] and pure) for 2 to 8 h while raising the temperature to 20 °C. Pure epoxy resin was added at room temperature. After polymerization 24 h at 60 °C, 60 to 80 nm sections were obtained using an ultra-microtome UC7 (Leica Microsystems) and an Ultra 35° diamond knife (DiATOME) and were collected on formvar-carbon-coated 100 mesh copper grids (EMS). The thin sections were post-stained for 10 min with 2% uranyl acetate, rinsed and incubated for 5 min with lead citrate. The samples were observed using a FEI Tecnai12 120 kV LaB6 microscope with an Orius SC1000 CCD camera (Gatan).

### Lipid extraction

Lipids were extracted from freeze-dried cells. First, cells were harvested by centrifugation and then immediately frozen in liquid nitrogen. Once freeze-dried, the pellet was suspended in 4 mL of boiling ethanol for 5 min to prevent lipid degradation and lipids were extracted according to Folch et al.[61] by addition of 2 mL methanol and 8 mL chloroform. The mixture was saturated with argon and stirred for 1 h at room temperature. After filtration through glass wool, cell remains were rinsed with 3 mL chloroform/methanol 2:1, v/v and 5 mL of NaCl 1% were added to initiate biphase formation. The chloroform phase was dried under argon and the lipid extract was stored at −20 °C.

### TLC and GC-FID/MS phospholipid quantification

Total phospholipids were quantified from their fatty acids. The lipid extract was solubilised in pure chloroform and 5 µg of C21:0 (internal standard) were added in an aliquot fraction. Fatty acids were converted to methyl esters (FAME) by a 1-hour incubation in 3 mL 2.5% H2SO4 in pure methanol at 100 °C[62]. The reaction was stopped by addition of 3 mL water and 3 mL hexane. The hexane phase was analyzed by gas chromatography coupled to flame ionisation detector (FID) and quadrupole mass spectrometer (full scan acquisition from 50 to 450 m/z) with an electronic impact source (Clarus 680 SQ8T MS, Perkin Elmer) on a BPX70 (SGE) column. FAMEs were identified by comparison of their retention times with those of standards (Bacterial Acid Methyl Ester (BAMEs) Mix, SigmaAldrich) and by their mass fragmentation spectra compared to the NIST (National Institute of Standards and Technology) database v4.2. They were quantified using C21:0 for calibration. To quantify each class of phospholipid, 300 µg of lipids were separated by two-dimensional thin layer chromatography (TLC) onto glass-backed silica gel plates (Merck)[63]. The first solvent was chloroform:methanol:water (65:25:4, v/v) and the second one chloroform:acetone:methanol:acetic acid:water (50:20:10:10:5, v/v). Lipids were sprayed with 2% 8-anilino-1-naphthalenesulfonic acid in methanol, then visualised under UV light and scraped off the plate. Lipids in the scraped silica were quantified by methanolysis and GC-FID/MS as described above. The total amount of PA in the *E. coli* membranes being extremely low, PA could not be detected in these experiments, meaning that in the conditions of this study PA represented less than 0.1% of the total membrane phospholipids.

### Dot blot assay (protein-lipid overlay assay)

All phospholipids were purchased from Avanti Lipids (Sigma). 2 µL of chloroform solutions (5, 0.5 and 0.05 mg/mL) of 1-palmitoyl-2-oleoyl-glycero-3-phosphocholine (16:0-18:1 PC), 1-palmitoyl-2-oleoyl-sn-glycero-3-phospho-L-serine (16:0-18:1 PS), 1-palmitoyl-2-oleoyl-sn-glycero-3-phosphoethanolamine (PE 16:0-18:1 PE), 1-palmitoyl-2-oleoyl-sn-glycero-3-phospho-(1'-rac-glycerol) (16:0-18:1 PG), 1-palmitoyl-2-oleoyl-sn-glycero-3-phosphate (16:0-18:1 PA) and 1',3'-bis[1-palmitoyl-2-oleoyl-sn-glycero-3-phospho]-

glycerol (16:0-18:1 Cardiolipin) were spotted using a Hamilton syringe onto a nitrocellulose membrane (Biorad trans-blot turbo RTA Midi 0.2 μm) to yield 10, 1 and 0.1 μg of the lipid per spot. The membranes were blocked using a Tris-buffered saline (TBS) solution (pH 7.4) supplemented with 10% (w/v) non-fat milk powder at room temperature for 1 h before incubated with 10 μg/mL of the purified proteins (His-RavA, AviTag-ViaA-His, LdcI-His) or protein complex (LdcI-RavA) in TBS containing 0.05 % (v/v) Tween-20 (TBST). LdcI was purified as previously described[14]. After overnight incubation at 4 °C, the membranes were washed with TBST (3 × 15 min) and probed either with a rat polyclonal antibody raised against the purified ViaA (Qalam-antibody) or a rabbit polyclonal antibody raised against the purified RavA or LdcI proteins (Qalam-antibody), diluted 1:10.000 in TBST, at room temperature for 1 h. The membranes were washed with TBST (3 × 15 min) and further incubated with either anti-mouse or anti-rabbit IgG secondary antibody conjugated with horseradish peroxidase (Merck), diluted 1:10.000 in TBST, at room temperature for 1 h. After washing with TBST (3 × 15 min), the chemiluminescence signal was developed using a Pierce ECL Western blotting substrate (Thermo Fisher Scientific, USA) and recorded using a ChemiDoc XRS + System (Bio-rad, USA).

For RavA bound via ViaA, the milk-blocked membrane was incubated with TBST alone (RavA only) or with 10 μg/mL AviTag-ViaA-His overnight at 4 °C, washed three times with TBST buffer and incubated with 10 μg/mL His-RavA for 2 h at room temperature. The chemiluminescent signal was developed after the incubation of the membranes with anti-RavA serum followed by incubation with the HRP-conjugated secondary antibody. Likewise, for ViaA bound via RavA, the membrane was incubated with 10 μg/mL His-RavA overnight before incubation with 10 μg/mL AviTag-ViaA-His. The LdcI-RavA complex was prepared by mixing the purified LdcI and His-RavA proteins in molar ratio of 1:2.25 (LdcI:RavA) before incubating for 30 min at room temperature. The milk-blocked membranes were incubated with His-RavA (10 μg/mL) or LdcI (6.5 μg/mL) alone or with the LdcI-RavA complex (corresponding to the final concentration of LdcI and RavA proteins of 6.5 μg/mL and 10 μg/mL, respectively) diluted in TBST supplemented with 10 mM MgCl₂ and 2 mM ADP. After overnight incubation at 4 °C, the membranes were washed and the chemiluminescent signal was developed after incubation of the membranes with anti-RavA or anti-LdcI sera followed by incubation with the HRP-conjugated secondary antibody (both diluted also in TBST-Mg-ADP buffer).

### Time-dependent killing assay
This experiment was performed as previously described[5]. Briefly, overnight cultures were diluted (1/100) and grown anaerobically at 37 °C to an $OD_{600}$ of 0.2 in LB medium supplemented with 10 mM fumarate. At this point (T0), Gm was added to the cells at 16 μg/mL. After 0, 1.5 and 3 h, 100 μL of cells were diluted in sterile phosphate buffered saline solution (PBS buffer), spotted on LB agar and incubated at 37 °C for 48 h. Cell survival was determined by counting colony-forming units per mL (CFU/mL). The absolute CFU at time-point zero (used as the 100%) was approximately $5 \times 10^7$ CFU/mL. This assay was performed in an anaerobic chamber (Jacomex, France).

### Complementation assay
To construct an MG1655-ΔviaA strain, the viaA deletion mutation from the KEIO collection was introduced by P1 transduction into the MG1655 strain. Transductant was verified by PCR, using a pair of primers hybridizing upstream (5′-GAAAGCGACTGGCGCAAGCAACAC-3′) and downstream (5′-GGCGGCGGTATCGGCCAGTCTCG-3′) of the viaA gene. The Kan resistance cassette was then removed using the pCP20 plasmid. For the complementation assay, the MG1655-ΔviaA strain was transformed with a low-copy auxiliary pT7pol26 (Kan^R) plasmid and the resulting MG1655-ΔviaA/pT7pol26 (Kan^R) strain was then

transformed with either an empty vector p∅, or a plasmid (Amp^R) carring His-PAmCherry-ViaA or His-ViaA-PamCherry construct under the control of the T7 promotor. The cells were cultured at 37 °C under anaerobic conditions in LB medium supplemented with fumarate at 10 mM, Kan at 50 μg/mL and Amp at 100 μg/mL. Overnight cultures were diluted 1/50 in LB medium supplemented with 10 mM fumarate and IPTG at 40 μM until the OD600 reached 0.2. At this point (T0), the cultures were divided into two halves and 16 μg/mL of Gm was added to one of the halves. After 0, 1.5 and 3 h, 100 μL of cells were diluted in sterile PBS buffer, spotted on LB agar and incubated at 37 °C for 48 h. Cell survival was determined by counting CFU/mL and normalized relative to T0. This assay was performed in an anaerobic chamber (Jacomex, France).

### Planar lipid bilayer experiments
Planar lipid bilayer measurements were performed at 25 °C using a home-made Teflon cells separated by a diaphragm with a circular hole with the diameter of 0.5 mm. The membrane was formed by a painting method using a mixture of lipids composed of PA:PG:PE (10:45:45) dissolved in n-decane and 3 % asolectin dissolved in n-decane:butanol (9:1) for ViaA and RavA proteins and the B. pertussis pore-forming CyaA toxin, respectively. The membrane current was recorded by Theta Ag/AgCl electrodes after addition of 2 nM AviTag-ViaA-His, His-RavA or 250 pM CyaA[64] in 10 mM Tris-HCl (pH 7.4), 0.15 M KCl and 2 mM CaCl₂ using applied potential of −50 and +50 mV. The signal was amplified by an LCA-200-100G Ultra-Low-Noise Current Amplifier (Femto, Germany) and digitised by a Keithley KPCI-3108 PCI Bus Data Acquisition Boards card. The signal was analysed using QuB (https://qub.mandelics.com) and processed using a 10-Hz filter.

### Reporting summary
Further information on research design is available in the Nature Research Reporting Summary linked to this article.

## Data availability
The data that support this study are available from the corresponding authors upon reasonable request. The source data underlying Figs. 1, 2, 4, 6, and Supplementary Figures 1, 5, 6, 7 and 8 are provided as a Source Data file. Source data are provided with this paper.

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

## Acknowledgements

We thank Alain Roussel and Aline Desmyter for production of anti-RavA and anti-ViaA nanobodies, and Ulrich Rothbauer and Ulrike Endesfelder for sharing their construct of the BC2-tag and their anti-BC2 nanobody with us. We are grateful to Megghane Baulard and Virgile Adam for help with initial construct design and optical imaging, to Beatrice Py for advice and training in generating chromosomal mutants, to Patricia Renesto for initial characterisation of AG sensitivity, to Humaira Khaliq for planar lipid bilayer measurements and to Dominique Bourgeois for discussions. This work was funded by the European Union's Horizon 2020 research and innovation programme under grant agreement No 647784 to IG. The nanobody generation platform of the AFMB laboratory (Marseille, France) was supported by the French Infrastructure for Integrated Structural Biology (FRISBI) ANR-10-INSB-05-01. We used the platforms of the Grenoble Instruct-ERIC center (ISBG; UAR 3518 CNRS-CEA-UGA-EMBL) within the Grenoble Partnership for Structural Biology (PSB), supported by FRISBI (ANR-10-INBS-0005-02) and GRAL, financed within the University Grenoble Alpes graduate school (Ecoles Universitaires de Recherche) CBH-EUR-GS (ANR-17-EURE-0003). The EM facility is supported by the Rhône-Alpes Region, the Fondation Recherche Medicale (FRM), the fonds FEDER and the GIS-Infrastrutures en Biologie Sante et Agronomie (IBISA). The LIPANG (Lipid analysis in Grenoble) hosted by the LPCV (UMR 5168 CNRS-CEA-INRAE-UGA) is supported by the Rhône-Alpes Region, the fonds FEDER, and GRAL, financed within the University Grenoble Alpes graduate school (Ecoles Universitaires de Recherche) CBH-EUR-GS (ANR-17-EURE-0003). J.F. was supported by a long-term European Molecular Biology Organization (EMBO) fellowship (ALTF441-2017) and a Marie Skłodowska-Curie actions individual fellowship (789385, RespViRALI). L.B. was supported by the National Institute of Virology and Bacteriology (Programme EXCELES, ID Project No. LX22NPO5103) - Funded by the European Union - Next Generation EU.

## Author contributions

J.F., L.B., C.L., A.F., F.R., J.Y.E.K., K.H., B.G., C.M., J.P.K., Y.D., M.J., E.K. and I.G. performed experiments, J.F., L.B., C.L., A.F., F.R., J.Y.E.K., J.P.K., Y.D., M.J., F.B., J.J. and I.G. analysed data. I.G. designed the overall study, supervised the project and wrote the paper with significant input from J.F. and contributions from L.B., C.L., A.F., F.R., K.H., J.P.K. and J.J. J.F. prepared the figures with contributions from L.B., C.L., A.F., F.R., J.Y.E.K., J.P.K., Y.D. and I.G. All authors edited the manuscript prior to submission.

## Competing interests

The authors declare no competing interests.

## Additional information

[1]Institut de Biologie Structurale, Univ Grenoble Alpes, CEA, CNRS, IBS, 71 Avenue des martyrs, Grenoble, France. [2]Institute of Microbiology, The Academy of Sciences of the Czech Republic, Videnska, 1083 Prague, Czech Republic. [3]Laboratoire de Physiologie Cellulaire Végétale, Univ Grenoble Alpes, CEA, CNRS, INRAE, IRIG, 17 Avenue des martyrs, Grenoble, France. [4]Institut Pasteur, Université de Paris, CNRS UMR6047, Stress Adaptation and Metabolism Unit, Department of Microbiology, Paris, France. [5]Univ Grenoble Alpes, CEA, CNRS, ISBG, 71 Avenue des martyrs, Grenoble, France. [6]Present address: Unit for Structural Biology, Department of Biochemistry and Microbiology, Ghent University, Ghent, Belgium. [7]Present address: Unit for Structural Biology, VIB-UGent Center for Inflammation Research, Ghent, Belgium. [8]Present address: EMBL Grenoble, 71 Avenue des martyrs, Grenoble, France. [9]Present address: Division of Structural Biology, The Institute of Cancer Research (ICR), London, UK. [10]Present address: European Synchrotron Radiation Facility, 71 Avenue des martyrs, Grenoble, France. [11]These authors contributed equally: Jan Felix, Ladislav Bumba, Clarissa Liesche. ✉e-mail: irina.gutsche@ibs.fr

