## [Peer Review File · Nature Communications]

The AAA+ ATPase RavA and its binding partner ViaA modulate E. coli aminoglycoside sensitivity through interaction with the inner membraneReviewers' Comments:

Reviewer #1:

Remarks to the Author:

The AAA+ ATPase RavA and its binding partner ViaA modulate E. coli aminoglycoside sensitivity through interaction with the inner membrane

Manuscript#: NCOMMS-22-08135A-Z

Corresponding author: Irina Gutsche

In this work authors have studied the interactions of the protein ViaA with the RavA ATPase and with phospholipids typical of the E. coli inner membrane. To do so, the protein VirA was cloned in fusion with different tags, as the over-expression of the wild type protein was unsatisfactory. The authors followed with the characterization of the chimeric proteins with a variety of techniques. First, the oligomeric state of the protein was determined by MALLS and SAXS, finding that the chimeric protein was mainly in a dimeric state. Although these experiments were well performed there a big concern is that if this dimeric state was due to the different tags added to ViaA. It is possible that the wild type protein, without any tag, may behave differently.

In a second part of the work, the authors investigated the interaction of Avitag-VirA-His protein with the well known RavA ATPase. As it is stated, in a first approach the authors were unable to form a stable complex between both proteins. This putative interaction was investigated in an indirect way, by Bio-Layer Interferometry. Using this approach the nucleotide binding properties of the mixture of both proteins was determined. These experiments showed clearly a preference of binding for ADP, but not for ATP. Taking into account these BLI results, it would have been interesting to repeat the VirA-RavA interaction experiments, previously failed, but this time in the presence of ADP. It would have also been interesting to show ATP hydrolysis experiments of RavA in the presence of VirA in which the ATPase activity was inhibited by ADP, to probe that the interaction between both proteins is functional.

The next chapter was on the localization of these proteins in E coli by optical and electron microscopy. The authors performed super-resolution microscopy, STORM and PALM, which are the state of the art in stochastic optical microscopy. One interesting result was to find RavA ATPase bound to the membrane, instead of being widely distributed along the cell cytoplasm. However, there is a possibility that this could be due to the fusion with a 12-residue long BC2 peptide. The conclusion of these experiments is that RavA localize at the inner membrane and that VirA modifies the morphology of the membrane, as the electro microscope images showed.

Following this membrane localization approach the authors studied the putative interaction of these proteins with E. coli phospholipids. They found that RavA binds PG and that VirA binds PA and CL. The binding sites in VarA for RavA and lipids was explored in 3D models of VirA protein obtained by Alphaphold and RosettaFold. These models were obtained using the sequence of VirA. However, all the experiments shown in this article were performed with chimeric forms of VirA in which different tags were included. In particular, all the experiments of lipid interactions and membrane localization of VirA were done with a chimeric protein of VirA, or its NTV and CTV domains, with PAmCherry. The problem here is that the authors tried to localize the lipid binding site in the 3D models of VirA. They should have created a 3D model of VirA-PAmCherry, which is the protein studied in the lipid interaction assays.

Finally, in the last part of this work, the authors studied the sensitivity to gentamicin of VirA and RavA mutants in fumarate respiratory conditions. They found that under these conditions the mutants exhibit an increased resistance to gentamicin, in comparison to wild type proteins. The results are very interesting.

Altogether, the work presented here is a compelling collection of a large variety of techniques to address the question of the role of VirA and its interaction with RavA and with lipids. Although the results are very informative, there is a hint of a doubt about some conclusions, considering that all the

experiments were not done with the wild type protein but with proteins fused to different tags. As it seems very difficult to obtain over-production of wild type VirA, the conclusions reached by the authors should not be so categorical, but they should be nuanced.

Therefore, unless the authors obtain wild type VirA protein the text should be modified to clearly state that all the results have been obtained with chimeric forms of the protein.

Reviewer #2:

Remarks to the Author:

In the present paper, the authors investigate the molecular basis of the aminoglycoside bactericidal activity under anaerobiosis in *E. coli*. The *ravA-viaA* operon plays a central role in this process and the objective of the paper is to uncover the functional interactions between RavA and ViaA proteins with *E. coli* inner membrane.

In figures 1 and 2, the authors convincingly show, using Sec-MALLS and SAXS methods that ViaA is a dimeric, soluble two-domain protein and that kinetics of the RavA-ViaA interaction are nucleotide-dependent. Then, they use high resolution fluorescent microscopy to demonstrate that, upon overexpression, both proteins show a propensity to localise at the cell periphery. Electron microscopy analysis revealed a close localization to the inner membrane and upon overexpression of ViaA, arrays of ectopic intracellular membrane tubes are observed. From this observation, the authors wonder if ViaA and RavA are able to bind lipids. Using a dot-blot approach, they show that purified RavA specifically binds PG, whereas ViaA specifically binds PA. Using *in silico* approaches and site directed mutagenesis they identified two charged amino-acids from ViaA, R476 and R479, that are responsible for the interaction with lipids. As expected, these mutants do not bind the inner membrane anymore and accumulate as soluble cytosolic bodies. In addition, overexpression of ViaA strongly increased the accumulation of cardiolipin (CL). In order to link the lipid binding feature of ViaA and RavA to gentamicin (Gm) sensitization, ViaA and RavA mutations were introduced in the chromosomal copy of the gene. As expected, all mutants exhibited increased resistance to Gm compared to the WT strain. In the discussion section, the authors conclude that ViaA-RavA do not act as respiration uncoupler but most likely as regulator of anionic phospholipid and CL in CL-enriched microdomains involved in proton sink and in the clustering of respiratory complexes.

The work presented here is technically sound and regulation of lipid microdomains by proteins in bacteria is a very exciting emerging field of research technically challenging to address. On the one hand the work presented here is new and clearly demonstrate that "RavA and its binding partner ViaA modulate *E. coli* aminoglycoside sensitivity through interaction with the inner membrane". On the other hand the molecular mechanism by which RavA/ViaA interact with the inner membrane is not clearly established.

I have the following comments:

1. Interaction between RavA and ViaA.

Figure 1 shows that ViaA is a dimeric protein and Figure 2 shows that RavA binds to ViaA. What is the stoichiometry of the complex ViaA-RavA? Is it 2/1?

2. RavA accumulates under the inner membrane whereas ViaA modifies membrane morphology of *E. coli* cells

The phenotype observed is upon over-expression of RavA or ViaA proteins, not with physiological concentrations. As mentioned by the authors several membrane proteins disturb membrane biogenesis and lipid homeostasis especially CL upon overproduction in *E. coli* (see Jamin et al. 2018, Royes et al. 2020). However, in most cases, these proteins expressed at *in vivo* (or physiological?) levels have no effect on membrane morphology and CL content.

What about the localisation of RavA and ViaA at physiological-like levels for instance under *ara* promoter with low concentrations of arabinose or under the T7 promoter but with a down-regulated T7

host ? Given that ViaA and RavA form a complex it would have been more appropriate to co-express both proteins to study their localization in *E. coli*. Regarding the arrays of ectopic intracellular membrane tubes observed upon overexpression of ViaA, they may not be related at all with gentamicin sensitisation and the *in vivo* mechanism of interaction of ViaA/RavA with the inner membrane. Figure 6 shows specific role of ViaA R476 and R479 but it does mean that the mechanism of action is through tubulation at inner membrane when the protein is present at physiological level.

3. Interaction of ViaA with CL.

Lane 207-212 (and 282): the authors state that ViaA interacts strongly with CL. However, Figure 2 A and Supplementary Figure 3 show, *in vitro*, no physical interaction with CL or PG. In Figure 3B all constructs increased the CL content, including ViaAR476R479, which is not supposed to bind CL or PA. At high concentration, ViaA certainly disturbs lipid homeostasis, perhaps by trapping PA (thanks to R476 and R479) which inhibits *clsA* enzyme, but the authors did not prove that the same mechanism applies at physiological, very low concentration.

Figure 6 shows an increase in gentamicin resistance with chromosomal copy, and therefore physiological levels of ViaA1-472, ViaAR476E/R479E and RavADLARA protein. Indeed, all three constructs, which are deficient in either PA or PG binding, play a crucial role in the mechanism of gentamicin resistance.

Given that the membrane remodeling activity of ViaA and RevA is observed at very high concentration, I wonder what is the gentamicin phenotype of RavA-OE, ViaA-OE and related mutants. To better link ViaA with CL metabolism, gentamicin resistance experiments may be performed in a *clsABC* deletion genetic context.

To conclude, the discussion is presently highly focused on CL metabolism, CL-enriched microdomain and CL-base proton sinks, which are both attractive given that gentamicin resistance is pmf dependent. However, this part of the discussion is poorly supported by the results. The authors have not shown a physical interaction between ViaA and CL and the lipid remodeling activity of both ViaA and RevA is not established at physiological expression levels.

I would therefore recommend to be more cautious in the discussion by focusing on results from Figures 2C, 4A, 5 and 6, which are fully in agreement with the title of the paper.

Reviewer #1 (Remarks to the Author):

We thank the reviewer for his overall positive assessment of our manuscript, and for his constructive comments. In the following paragraphs we will answer in detail on each of the following remarks.

In this work authors have studied the interactions of the protein ViaA with the RavA ATPase and with phospholipids typical of the *E. coli* inner membrane. To do so, the protein ViaA was cloned in fusion with different tags, as the over-expression of the wild type protein was unsatisfactory.

The authors followed with the characterization of the chimeric proteins with a variety of techniques. First, the oligomeric state of the protein was determined by MALLS and SAXS, finding that the chimeric protein was mainly in a dimeric state. Although these experiments were well performed there a big concern is that if this dimeric state was due to the different tags added to ViaA. It is possible that the wild type protein, without any tag, may behave differently.

While it is possible in our hands to express Wild-Type (His-tagged) ViaA, the yield is indeed too low for subsequent biophysical studies such as multi-angle laser light scattering (MALLS) or small-angle X-Ray scattering (SAXS). Furthermore, with this construct, substantial degradation was present after a final size-exclusion chromatography (SEC) purification. While cloning and expressing ViaA constructs that contained either an N- or C-terminal AviTag for attachment to the BLI sensor, we observed that the construct with an N-terminal AviTag displayed a 10-fold increase in yield, presumably due to protection of the N-terminus from degradation, and ran as a far more monodisperse peak after SEC (see Figure 1 added below). This is supposedly due to the fact that the first residue after the first Methionine in WT ViaA is Leucine (5.5 hour half-life according to the N-end rule) while it is a Glycine (30 hour half-life) for ViaA with an N-terminal AviTag.

While constructs with bigger tags were used for subsequent PALM and STORM experiments (whereas the wide-field imaging was performed with His-tagged only proteins because they could be directly stained with fluorescently labelled anti-RavA and anti-ViaA nanobodies), all MALLS and SAXS studies were performed with a ViaA construct with an N-terminal AviTag and C-terminal His tag. While, as published by us (Jessop et al., PNAS 2020) and by others, some tags are indeed known to induce oligomerization, the small and flexible tags on the AviTag-ViaA-His construct are highly unlikely to cause any dimerization. Furthermore, ViaA constructs with an N- or C-terminal AviTag elute at the same elution volume on a SD200increase SEC column, demonstrating that the dimerization of ViaA is independent of the position of the AviTag (Figure 1).

Figure 1: SEC elution profile of AviTag-ViaA-His (green curve) and His-ViaA-AviTag (blue curve) constructs injected onto a SD200increase column.

To clarify this, we have now added the following to the Manuscript:

Line 81: “Initial attempts to purify a HIS-tagged ViaA construct using immobilized metal affinity chromatography (IMAC) and size-exclusion chromatography (SEC) (Materials and Methods) resulted in prohibitively low amounts of purified ViaA protein. Addition of an N-terminal AviTag to C-terminally His-tagged ViaA (hereafter named AviTag-ViaA-His) stabilized the protein and enabled its high yield purification for characterisation by multi-angle laser light scattering (MALLS) and small-angle X-ray scattering (SAXS) (Figure 1B-D, Materials and Methods).

Line 94: “We note that His-ViaA-AviTag and AviTag-ViaA-His constructs elute at the same volume after SEC, and that the elongated character and apparent flexibility of ViaA might be accentuated by the presence of non-cleaved tags.”

In a second part of the work, the authors investigated the interaction of Avitag-ViaA-His protein with the well known RavA ATPase. As it is stated, in a first approach the authors were unable to form a stable complex between both proteins.

Indeed, addition of pre-purified RavA to ViaA prior to injection onto a SD200increase SEC column does not result in the formation of a stable complex. However, this is rather unsurprising given that the affinity between ViaA and RavA observed via BLI interaction studies is in the micromolar range. This low affinity seems to be an intrinsic biological property of these proteins and prevents formation of a stable complex via SEC. This explains why pull-down experiments with SPA-tagged RavA don't bring down ViaA and vice versa (Wong et al., JMB 2017) and why previous determinations of the dissociation constant of the RavA-ViaA complex could only be performed indirectly, by observing the effect of ViaA on the ATPase activity of RavA (Wong et al., JMB 2017).

This putative interaction was investigated in an indirect way, by Bio-Layer Interferometry.

We disagree that BLI is an indirect way of measuring protein-protein interactions. In contrast to the previous indirect observations based on the variations of the RavA ATPase activity in presence of the increasing concentrations of ViaA (Snider et al., JBC 2006; Wong et al., JMB 2017), in this work we provide the first direct measurement of the strength of the RavA-ViaA interaction. Indeed, BLI is a direct label-free method, similar to Surface Plasmon Resonance (SPR) and Isothermal Titration Calorimetry (ITC). A downside of BLI is the immobilization of one of the binding partners onto a BLI sensor tip, which can result in orientation effects. To overcome this, we cloned ViaA with both an N- and C- terminal AviTag for mobilization on Streptavidin-tips and observed negligible differences in the interaction with RavA.

Using this approach the nucleotide binding properties of the mixture of both proteins was determined. These experiments showed clearly a preference of binding for ADP, but not for ATP. Taking into account these BLI results, it would have been interesting to repeat the ViaA-RavA interaction experiments, previously failed, but this time in the presence of ADP.

As explained above, the observation by us and by others (Snider et al., JBC 2006; Wong et al., JMB 2017) that the RavA-ViaA complex is not stable enough for the two proteins to co-elute from a SEC column, does not mean that the experiments fail but corroborates the data showing the micromolar affinity between RavA and ViaA.

We tried to repeat the SEC experiments with RavA + ADP, and supplemented ADP in the SEC buffer, but this did not result in a stable ViaA-RavA complex (data not shown). Even after addition of ADP (or ATP_γS or AMPPNP), the affinity is still too low to result in a stable complex via SEC.

It would have also been interesting to show ATP hydrolysis experiments of RavA in the presence of ViaA in which the ATPase activity was inhibited by ADP, to probe that the interaction between both proteins is functional

As can be seen from the performed BLI experiments (Figure 2), the mode of interaction between ViaA and RavA seems to be different with ADP/ATP. However, this does not mean that the interaction becomes less functional. ATP hydrolysis causes conformational changes in RavA (Jessop et al., 2020), which seem to have an influence on the shape of the RavA-ViaA binding curve. Yet, binding is

clearly present after adding ADP or ATP. The fact that the binding curves of the ViaA-RavA interaction after addition of ATP could only be fitted using a heterogeneous ligand binding model may point to the presence of a mixture of RavA conformations after addition of ATP, with different ViaA binding affinities.

The ATPase activity of RavA in the presence of ViaA was previously investigated by others (Snider *et al.*, JBC 2006; Wong *et al.*, JMB 2017) and these studies are cited in our manuscript.

We have now added following sentence to the manuscript for clarification:

Line 110: “ADP binding to RavA increased the strength of the RavA-ViaA interaction to $\sim 2.2 \mu\text{M}$, whereas the presence of ATP altered the shape of the BLI curves that could only be reliably fitted using a 2:1 heterogeneous ligand binding model, resulting in $KD1/KD2$ values of $30.3/1.0 \mu\text{M}$ and $37.5/0.5 \mu\text{M}$ for His-ViaA-AviTag and AviTag-ViaA-His respectively. The latter may point to the presence of a mixture of RavA conformations after addition of ATP, with different ViaA binding affinities.”

The next chapter was on the localization of these proteins in *E. coli* by optical and electron microscopy. The authors performed super-resolution microscopy, STORM and PALM, which are the state of the art in stochastic optical microscopy. One interesting result was to find RavA ATPase bound to the membrane, instead of being widely distributed along the cell cytoplasm. However, there is a possibility that this could be due to the fusion with a 12-residue long BC2 peptide. The conclusion of these experiments is that RavA localizes at the inner membrane and that ViaA modifies the morphology of the membrane, as the electron microscopy images showed.

The original papers describing the short BC2-tag (Braun *et al.*, 2016, Virant *et al.* 2020) show that this tag is inert and has no effect on the biological properties of the tagged constructs. Furthermore, given the fact that BC2-tag labeling is based on the tight binding of a fluorescently labeled nanobody to the BC2-tag, we would assume that if the binding of BC2-tagged RavA would be mediated by the BC2-tag itself, it would not be available for binding with the fluorescently labeled nanobody in the first place, which would result in an absence of fluorescent signal. Finally, as shown in Supplementary Figure 2 of our manuscript, we used an anti-RavA nanobody for wide field imaging of the overexpressed His-RavA, which shows that the same distribution is found as for BC2-tagged RavA.

Following this membrane localization approach the authors studied the putative interaction of these proteins with *E. coli* phospholipids. They found that RavA binds PG and that ViaA binds PA and CL. The binding sites in VarA for RavA and lipids was explored in 3D models of ViaA protein obtained by AlphaFold and RosettaFold. These models were obtained using the sequence of ViaA. However, all the experiments shown in this article were performed with chimeric forms of ViaA in which different tags were included. In particular, all the experiments of lipid interactions and membrane localization of ViaA were done with a chimeric protein of ViaA, or its NTV and CTV domains, with PAmCherry. The problem here is that the authors tried to localize the lipid binding site in the 3D models of ViaA. They should have created a 3D model of ViaA-PAmCherry, which is the protein studied in the lipid interaction assays.

The PAmCherry fusions of ViaA were only used for PALM imaging experiments (e.g. in Figure 3 and in the Supplementary Figure 2A). The dot-blot assays were performed with (Avi-)His tagged proteins. We realize this was not clearly stated in the Methods section and we have now adapted the entire Methods section accordingly to make clear which construct was used for which study. It is very unlikely that the addition of a single (Avi-)His tag will have any effect on the specific lipid binding properties of the respective proteins.

It should also be noted that the lipid binding-deficient mutant versions of RavA and ViaA that we created in this work also contain AviTag and/or His-tags. Since these mutants lose interaction with lipids although they do contain tags, this strongly supports the argument that the tags themselves do not have any propensity of binding lipids. Again, thanks to the reviewer's comments we realize that this was currently not entirely clear from the Methods section, and we have now updated the relevant sections accordingly.

Finally, regarding ViaA-PAmCherry, we would like to note that this construct has the same function as the WT ViaA, as can be seen from a complementation assay which is now added to the manuscript (Supplementary Figure 7). A Δ ViaA strain supplemented with a ViaA-PAmCherry or PAmCherry-ViaA plasmid, has the same phenotype as a WT *E. coli* strain after addition of Gentamycin.

Finally, in the last part of this work, the authors studied the sensitivity to gentamicin of ViaA and RavA mutants in fumarate respiratory conditions. They found that under these conditions the mutants exhibit an increased resistance to gentamicin, in comparison to wild type proteins. The results are very interesting.

We thank the review for highlighting the importance of this experiment in linking the RavA and ViaA membrane binding properties to their effect on the aminoglycoside sensitisation. We hope that the addition of the Supplementary Figure 8 with the complementation assay showing the ability of the PAmCherry fusion constructs to restore the sensitisation phenotype lifts the doubts regarding the effect of the tags on the observations described in our manuscript.

Altogether, the work presented here is a compelling collection of a large variety of techniques to address the question of the role of ViaA and its interaction with RavA and with lipids. Although the results are very informative, there is a hint of a doubt about some conclusions, considering that all the experiments were not done with the wild type protein but with proteins fused to different tags. As it seems very difficult to obtain over-production of wild type ViaA, the conclusions reached by the authors should not be so categoric, but they should be nuanced. Therefore, unless the authors obtain wild type ViaA protein the text should be modified to clearly state that all the results have been obtained with chimeric forms of the protein.

Based on the comments of reviewer 1, we realise that it is not always clear in our manuscript which tagged RavA/ViaA construct were used for the different experiments displayed in our study, and that the previous Supplementary Table 1 that was supposed to serve this purpose was not informative enough. We now split this Supplementary Table in two (numbered 2 and 3), and most importantly, we now provide an additional Supplementary Table 1 with all constructs used, the exact nature of the tag and the experiments they were used for.

We have addressed all concerns of Reviewer 1 regarding specific tags in detail in the paragraphs above, supplemented with a few extra experimental display items, and have added clarifications to the relevant Methods sections. We hope this is sufficient to answer this final comment.

Reviewer #2 (Remarks to the Author):

In the present paper, the authors investigate the molecular basis of the aminoglycoside bactericidal activity under anaerobiosis in *E. coli*. The *ravA-viaA* operon plays a central role in this process and the objective of the paper is to uncover the functional interactions between RavA and ViaA proteins with *E. coli* inner membrane. In figures 1 and 2, the authors convincingly show, using Sec-MALLS and SAXS methods that ViaA is a dimeric, soluble two-domain protein and that kinetics of the RavA-ViaA interaction are nucleotide-dependent. Then, they use high resolution fluorescent microscopy to demonstrate that, upon overexpression, both proteins show a propensity to localise at the cell periphery. Electron microscopy analysis revealed a close localization to the inner membrane and upon overexpression of ViaA, arrays of ectopic intracellular membrane tubes are observed. From this observation, the authors wonder if ViaA and RavA are able to bind lipids. Using a dot-blot approach, they show that purified RavA specifically binds PG, whereas ViaA specifically binds PA. Using *in silico* approaches and site directed mutagenesis they identified two charged amino-acids from ViaA, R476 and R479, that are responsible for the interaction with lipids. As expected, these mutants do not bind the inner membrane anymore and accumulate as soluble cytosolic bodies. In addition, overexpression of ViaA strongly increased the accumulation of cardiolipin (CL). In order to link the lipid binding feature of ViaA and RavA to gentamicin (Gm) sensitization, ViaA and RavA mutations were introduced in the chromosomal copy of the gene. As expected, all mutants exhibited increased resistance to Gm compared to the WT strain. In the discussion section, the authors conclude that ViaA-RavA do not act as respiration uncoupler but most likely as regulator of anionic

phospholipid and CL in CL-enriched microdomains involved in proton sink and in the clustering of respiratory complexes.

The work presented here is technically sound and regulation of lipid microdomains by proteins in bacteria is a very exciting emerging field of research technically challenging to address. On the one hand the work presented here is new and clearly demonstrate that “RavA and its binding partner ViaA modulate *E. coli* aminoglycoside sensitivity through interaction with the inner membrane”. On the other hand the molecular mechanism by which RavA/ViaA interact with the inner membrane is not clearly established.

We are grateful to the reviewer for the in-depth understanding of our work and for the recognition of the novelty and value it provides. We entirely agree that the main focus of the present manuscript is to demonstrate the direct interaction of enterobacterial RavA and ViaA with specific phospholipids in the *E. coli* inner membrane, elucidate the molecular determinants of this interaction and disclose its effect on the *E. coli* aminoglycoside sensitivity under anaerobiosis. While this work does not yet shed full light on the exact molecular mechanism of the RavA-ViaA function, it enables us to propose sound data-driven hypotheses that should stimulate further research in this area. Indeed, although the enterobacterial *ravAviaA* operon is being intensely studied since more than 15 years, the mechanism of RavA/ViaA action in bacterial physiology has so far been completely unknown.

I have the following comments:

1. Interaction between RavA and ViaA. Figure 1 shows that ViaA is a dimeric protein and Figure 2 shows that RavA binds to ViaA. What is the stoichiometry of the complex ViaA-RavA? Is it 2/1?

The inherently low expression and purification yield of ViaA constructs did not allow us to perform ITC experiments, which may provide a direct answer to the stoichiometry of the interaction. However, recent cryo-EM studies of the ribosome maturation factor Rea1 (that possesses both an AAA+ domain and a VWA domain inside the same subunit) shows that the VWA domain of one of the six subunits docks in the center of the AAA+ ATPase ring (Sosnowski et al., eLife 2018; Chen et al., Cell 2018). In addition, a low resolution negative stain EM map of the Rubisco-activating complex between the *A. ferrooxidans* MoxR-related protein CbbQ and the VWA domain-containing CbbO also seems to indicate occlusion of the center of the CbbQ ring by a VWA domain of CbbO (Tsai et al., PNAS 2020). Therefore, we speculate that one RavA hexamer is likely to bind a VWA domain of one ViaA protein (monomer or dimer). While we are now actively working on the stabilisation of the RavA-ViaA complex in view of its high resolution structural analysis by cryo-EM, this ongoing work is beyond the scope of this study and therefore we would prefer not to discuss the stoichiometry of the RavA-ViaA binding in the present manuscript.

2. RavA accumulates under the inner membrane whereas ViaA modifies membrane morphology of *E. coli* cells. The phenotype observed is upon over-expression of RavA or ViaA proteins, not with physiological concentrations. As mentioned by the authors several membrane proteins disturb membrane biogenesis and lipid homeostasis especially CL upon overproduction in *E. coli* (see Jamin et al. 2018, Royes et al. 2020). However, in most cases, these proteins expressed at in vivo (or physiological?) levels have no effect on membrane morphology and CL content.

We fully agree with this comment of the reviewer. However, we think that several important points should be considered here.

First, before this study, ViaA (and RavA) was not even known to interact with membranes. The present manuscript is the first to show that ViaA is actually a peripheral membrane protein tethered to the *E. coli* inner membrane via an amphipathic helix and to identify its lipid binding site as being strikingly similar to CL-binding sites of integral membrane proteins as very recently determined by molecular dynamics simulations (Corey et al., Science Advances 2021).

Second, we directly show that mutant RavA and ViaA constructs that are unable to interact with specific lipids in a dot-blot assay set-up, do not considerably modify the CL content and don't induce formation of intracellular membrane proliferation when overexpressed.

Third, as summarised in the Table 1 of the excellent review by Royes et al., (Microbial Cell Factories 2020), only a dozen of proteins, very different from ViaA and described as “a few peculiar cases” have so far been described as triggering membrane proliferation in prokaryotes.

Fourth, most of the proteins known to trigger formation of inducible intracellular membranes are also known to create zones with high membrane curvature and affect both membrane morphology and phospholipid biosynthesis. As we propose in the discussion section, when expressed at natively low levels, ViaA may act on certain membrane properties, such as membrane curvature, locally. This hypothesis remains to be verified but is in our view very important to mention.

We realise that the review of Royes et al. 2020 was not cited in the first version of the manuscript. We now add this important citation that actually contributed crucial information and ideas for interpretation of our results.

What about the localisation of RavA and ViaA at physiological-like levels for instance under ara promoter with low concentrations of arabinose or under the T7 promotor but with a down-regulated T7 host ?

To our knowledge, the only previous description of an attempt to localise RavA and ViaA at physiological-like levels has so far been provided by Boyu Zhao from the Walid Houry lab in the University of Toronto in the frame of her Master’s thesis research (<https://tspace.library.utoronto.ca/handle/1807/33622>). In this work, endogenous RavA and ViaA were C-terminally tagged with RFP and YFP respectively. While Western blot analysis showed that the fusion proteins were expressed under imaging conditions, no fluorescence signal could be observed, which was tentatively explained by a low expression level of both proteins in the conditions tested (at most ~350 RavA and ~100 ViaA molecules per cell, see Wong et al., PLoS One, 2014).

We are very aware of the necessity of further advanced imaging studies of RavA and ViaA at closer to physiological conditions. This will indeed be one of the objectives of future research in our lab, where we will also address potential co-localisation of RavA, ViaA and LdcI, as well as CL and respiratory complexes. Our preliminary observations of ViaA-mCherry expressed at a near-physiological level indicate a patchy peripheral distribution (Figure 2).

Figure 2. PALM imaging of a chromosomal MG1655 mutant strain harboring a ViaA-PamCherry fusion.

Excitingly, this distribution is remarkably similar to the one that we recently documented for the endogenous LdcI (Jessop et al., PNAS 2021) and seems therefore to provide a further link with CL-enriched membrane microdomains that compartmentalise respiratory complexes. However, these experiments need to be repeated and a more extensive and technically challenging 3D imaging study conducted. This, in the frame of the current manuscript, we decided to present the localisations of the overexpressed RavA and ViaA only, and to introduce a possibility of RavA and ViaA action on lipid microdomains and respiratory complexes as an attractive hypothesis that deserves further in-depth investigation.

3. Given that ViaA and RavA form a complex it would have been more appropriate to co-express both proteins to study their localization in *E. coli*.

We fully agree and are indeed planning to perform these studies in the future (see answer above).

4. Regarding the arrays of ectopic intracellular membrane tubes observed upon overexpression of ViaA, they may not be related at all with gentamicin sensitisation and the in vivo mechanism of interaction of ViaA/RavA with the inner membrane. Figure 6 shows specific role of ViaA R476 and R479 but it does mean that the mechanism of action is through tubulation at inner membrane when the protein is present at physiological level.

We completely agree that the observed intracellular membrane tubulation are a result of a (non-physiological) overexpression of ViaA. However, the membrane binding properties of ViaA demonstrated in our manuscript may locally induce/alter the membrane curvature or locally change some other membrane properties (such as for instance its permeability, fluidity, etc). The observations that (i) the overexpressed ViaA mutants deficient in lipid binding are also deficient in membrane localisation and tubulation and induce a much milder increase in the cellular CL content, and that (ii) the chromosomal lipid binding-deficient ViaA mutants exhibit increased resistance to aminoglycosides leads us to propose a mechanistic link between the lipid binding properties of ViaA and its effect on the aminoglycoside sensitisation. However, we are very careful not to claim that this mechanism involves tubulation and simply propose that it involves local changes of membrane properties.

5. Lane 207-212 (and 282): the authors state that ViaA interacts strongly with CL. However, Figure 2 A and Supplementary Figure 3 show, in vitro, no physical interaction with CL or PG.

We kindly disagree with the reviewer regarding this crucial point and suppose that this misinterpretation of the dot-blot assay figures (Figure 4 and Supplementary Figure 3) is indeed what may have given rise to the further divergence of our points of view regarding a potential link of ViaA with CL microdomains, proton sinks and respiratory complexes. We suppose that this interpretation of our results by the reviewer comes from the prominent difference in intensity between the hugely intense PA spot and the much less intense CL spot: the markedly intense PA spot makes indeed the CL spot underneath barely visible. Noteworthy, the relative amount of PA in the *E. coli* membranes is extremely low in comparison to CL. In our lipid quantification experiments, the amount of PA was below the detection level, i.e. less than 0.1% of total membrane phospholipids.

We would like to specify that a dot-blot assay is not a quantitative but rather a semiquantitative method. Therefore, this assay cannot be used to produce calibration curves and to infer binding affinity constants, and the intensities among individual dots should not actually be compared. Importantly, in this assay the lipids are not in a physiological state but are adsorbed to the membrane after evaporation of the solvent, which means that the “orientation” of the different lipids and their capacity to interact with the proteins may not be comparable. The CL spots are however clearly present in the ViaA panel, similar to the anti-ViaA nanobody control (at both 10 and 1 microgram of lipid/spot), showing that both CL and the anti-ViaA nanobody do interact with ViaA whereas for example PG hardly interacts and PE, PS and PC (the latter two used as negative controls and absent in MG1655) don't show any interaction at all. While the PA spot greatly decreases in the R472A/R479A mutant, the CL and the nanobody spots remain nearly unaltered by this mutation. However, both the ViaA₁₋₄₇₂ and the R472E/R479E mutants don't show any notable spots which means that these mutations abrogate the PA and the CL binding properties of ViaA.

Finally, we would like to stress that the lipid binding site identified experimentally and mapped on the AlphaFold/RoseTTAFold model in this work is strikingly similar to the one inferred by MD simulations on *E. coli* CL binding proteins (Corey et al., Science Advances 2021).

For clarity, we added following sentence to the Manuscript:

Line 187: “To investigate whether ViaA and RavA are capable of directly binding to lipids, we employed dot-blot assays using PE, PG, CL, PA as well as phosphatidylcholine (PC) and phosphatidylserine (PS) (see Material and Methods). While dot-blot assays are a semiquantitative method, and no direct comparison can be made between intensities of individual dots, they are nonetheless indicative of the relative propensity of a protein to interact with a particular lipid immobilised on a nitrocellulose membrane.”

6. In Figure 3B all constructs increased the CL content, including ViaAR476R479, which is not supposed to bind CL or PA.

Indeed we noticed this as well and specifically mention this observation in the corresponding results section (“As expected from the electron microscopy observations, the most spectacular changes were observed with CL, the proportion of which was 6 to 10-fold increased in the ViaA-OE cells, whereas overexpression of the other constructs led to a much milder (but still statistically significant) CL increase in comparison to the WT bacteria (Figure 4B)”). The amplitude of the effects is clearly very different and in our view simply means that the mutants do not completely lose their effect on lipid homeostasis.

In addition, as we also note in the main text, “As shown in Supplementary Figure 6, RavA-OE, RavA_{ΔLARA}-OE, ViaA_{R476E/R479E}-OE and ViaA₁₋₄₇₂-OE bacteria contained slightly but statistically significantly fewer lipids, quantified by their fatty acids, than the wild type (WT) MG1655 cells, whereas the total fatty acid content in ViaA-OE remained roughly similar to the WT.” This observation again points to an effect of the overexpressed WT and mutant RavA and ViaA on lipid homeostasis.

7. At high concentration, ViaA certainly disturbs lipid homeostasis, perhaps by trapping PA (thanks to R476 and R479) which inhibits *clsA* enzyme, but the authors did not prove that the same mechanism applies at physiological, very low concentration.

Indeed, we formulate this hypothesis in the discussion section of the manuscript: “A possible explanation of the effect of ViaA on CL can be proposed based on our finding that ViaA interacts with PA (Figure 4A). Indeed, PA, PG and CL are tightly interconnected: PA, only present in tiny amounts in *E. coli* cells, is a metabolic hub for PG synthesis, whereas two PG molecules are necessary to produce CL⁴¹. The synthesis of CL in bacteria requires that a first PG receives a phosphatidyl group from a second PG by a transesterification reaction catalysed by cardiolipin synthase (Cls). *E. coli* contains three Cls isoforms⁴², with ClsA being the major contributor to CL synthesis during exponential growth. Interestingly, PA was shown to strongly inhibit ClsA activity⁴³. Consequently, the association of ViaA with PA at the inner membrane level could trap PA, thereby preventing Cls inhibition and eventually favouring CL synthesis, leading to a modification in the lipid homeostasis.” Here, we do not claim that the same mechanism applies at physiological concentrations of ViaA – the goal of this paragraph is rather to explain why we observe a CL increase in our setup. Moreover, the sentence following immediately after, admits that this explanation may not be the only one and that anyway, at the native expression levels, ViaA would only act locally: “Independently of the concrete mechanism, one may anticipate that by binding to PA and CL, natively expressed ViaA would influence some local membrane properties...”

Figure 6 shows an increase in gentamicin resistance with chromosomal copy, and therefore physiological levels of ViaA1-472, ViaAR476E/R479E and RavADLARA protein. Indeed, all three constructs, which are deficient in either PA or PG binding, play a crucial role in the mechanism of gentamicin resistance. Given that the membrane remodeling activity of ViaA and RevA is observed at very high concentration, I wonder what is the gentamicin phenotype of RavA-OE, ViaA-OE and related mutants.

Co-authors of our present manuscript, Jessica El Khoury and Frédéric Barras, conducted a thorough analysis of the role of *ravA-viaA* in aminoglycoside sensitisation (El Khoury et al., bioRxiv 2021) and clearly demonstrated that overexpression of RavA-ViaA was required to sensitise *E. coli* to gentamicin in the presence of oxygen, whereas in the anaerobic conditions, with no exogenous electron acceptor or upon fumarate respiration, the sensitisation phenotype does not require RavA-ViaA overexpression. This agrees with the previous publication (Wong et al., JMB 2017) showing that the *ravA-viaA* operon was regulated by the anaerobic sensing transcriptional regulator Fnr. Thus, in the present study we choose not to focus on the effect of overexpression of the lipid binding-deficient RavA and ViaA mutant on the gentamicin phenotype, but to construct lipid binding-deficient chromosomal mutants of *ravA* and *viaA* and to address their gentamicin phenotype with physiological gene dosage.

To better link ViaA with CL metabolism, gentamicin resistance experiments may be performed in a *clsABC* deletion genetic context.

We thank the reviewer for this excellent suggestion. We have now created a MG1655-deltaCLS strain and are in process of constructing all combinations of mutations (in *cls* genes alone or in combination with *ravA*, *viaA* or both). This will provide us with an exhaustive set of combinations which we will test for Gm phenotype under different respiratory conditions. The collection of these new data should enable us to advance towards the mechanistic understanding of the *ravA-viaA* function but will clearly be a new paper.

To conclude, the discussion is presently highly focused on CL metabolism, CL-enriched microdomain and CL-base proton sinks, which are both attractive given that gentamicin resistance is pmf dependent. However, this part of the discussion is poorly supported by the results. The authors have not shown a physical interaction between ViaA and CL and the lipid remodeling activity of both ViaA and RevA is not established at physiological expression levels. I would therefore recommend to be more cautious in the discussion by focusing on results from Figures 2C, 4A, 5 and 6, which are fully in agreement with the title of the paper.

We have now shortened the discussion and specify that we anticipate that membrane remodelling activity of ViaA would only be exerted locally.

We now also pay attention to carefully specify in the text when phenotypes were observed with overexpressed proteins and nuance our interpretations where required.

However, as explained above, our dot-bot assays do show a physical interaction between ViaA and CL, and we believe this interaction to be important. We fully agree with the Reviewer that the current work does not give a formal proof that natively expressed ViaA remodels the membranes or that it comes into close proximity to CL-enriched lipid microdomains that host respiratory complexes. Nevertheless, we feel that the goal of the discussion section is precisely to make hypotheses based on the results presented in the manuscript and critically evaluate these in the context of the available literature.

Reviewers' Comments:

Reviewer #1:

Remarks to the Author:

Comments to the responses by authors to Reviewer_1's remarks

The authors have addressed most of the remarks. The main concern of this reviewer was related to the fact that most of the results shown in this manuscript were obtained with chimeric forms of the protein VirA, and, therefore, somehow, the physiological relevance of them were questionable. The work would have been more relevant if the same experiments was been done with the wild type protein. Unfortunately, they were unable to obtain high yields of purified wt VirA. In particular most of the experiments were done with an Avi-tag and a His tag at the N and C termini of VirA, respectively. The authors claim that these tags do not affect the oligomeric state of the protein. However, this is impossible to know in the absence of data with wild type protein. They tried to overcome this problem by comparing the size exclusion chromatography profiles of Avi-VirA-His with His-VirA-Avi constructs. Unfortunately, the profile obtained with the second construct was very poor and no conclusions can be obtained from this. The authors have added two sentences at lines 81 and 94. However, because of the messy profile of the His-VirA-Avi construct the sentence added in line 94 is not accurate.

Regarding the low affinity of the interactions of VirA with RavA ATPase, the authors mention that this is an intrinsic biological property of the interaction between these proteins. In fact, previous works (Snider et al., JBC 2006; Wong et al., JMB 2017) showed that the ATPase activity of RavA was not altered by addition of VirA. The authors claim that their approach, based on BLI, was more direct and thanks to this technique now they are able to measure interactions, albeit at micromolar range. In any case it seems obvious that this interaction is not stable enough to keep them together as a complex in a SEC column. Then, they followed to see if the ATPase activity of RavA was affected by VirA. The results were explained as if different conformational states of RavA in the presence of ATP or ADP were affecting the interaction of RavA with VirA. This makes sense but, in physiological conditions, to which RavA conformation VirA binds?

Regarding the remarks about the localization of these proteins as observed by microscopy, the explanations of the authors seem convincing.

In general, the authors have answered to most of the concerns, adding new data (Supplementary Table I and Supplementary Figure 8) and splitting the previous Supp Table I in two (Supp. Tables 2 and 3). They also add more information in the Methods Section.

Reviewer #2:

Remarks to the Author:

The authors convincingly replied to my concerns, I have no further comments or request.

Reviewer #1 (Remarks to the Author):

We thank the reviewer for the constructive and critical evaluation of our manuscript, and have provided answers to the few remaining remarks as shown below.

The authors have addressed most of the remarks. The main concern of this reviewer was related to the fact that most of the results shown in this manuscript were obtained with chimeric forms of the protein VirA, and, therefore, somehow, the physiological relevance of them were questionable. The work would have been more relevant if the same experiments were done with the wild type protein. Unfortunately, they were unable to obtain high yields of purified wt VirA. In particular most of the experiments were done with an Avi-tag and a His tag at the N and C termini of VirA, respectively. The authors claim that these tags do not affect the oligomeric state of the protein. However, this is impossible to know in the absence of data with wild type protein. They tried to overcome this problem by comparing the size exclusion chromatography profiles of Avi-VirA-His with His-VirA-Avi constructs. Unfortunately, the profile obtained with the second construct was very poor and no conclusions can be obtained from this. The authors have added two sentences at lines 81 and 94. However, because of the messy profile of the His-VirA-Avi construct the sentence added in line 94 is not accurate.

Although we interpret the elution time of the His-ViaA-AviTag construct as being indicative of a dimer, we agree that the low signal and poor quality of the SEC profile does not provide a conclusive answer. Ideally, a MALLS experiment could have provided a solution, but the very poor yield of the His-ViaA-AviTag construct does not allow for this. We propose to remove the sentence added in line 94 (“We note that His-ViaA-AviTag and AviTag-ViaA-His constructs elute at the same volume after SEC”) and add the following sentence at the end of the respective paragraph:

“Although the dimeric character of the purified AviTag-ViaA-His construct was verified by both MALLS and SAXS, we cannot exclude the possibility that non-tagged ViaA may harbor a different stoichiometry in solution. Furthermore, we note that the elongated character and apparent flexibility of ViaA might be accentuated by the presence of non-cleaved tags”

Regarding the low affinity of the interactions of VirA with RavA ATPase, the authors mention that this is an intrinsic biological property of the interaction between these proteins. In fact, previous works (Snider et al., JBC 2006; Wong et al., JMB 2017) showed that the ATPase activity of RavA was not altered by addition of VirA. The authors claim that their approach, based on BLI, was more direct and thanks to this technique now they are able to measure interactions, albeit at micromolar range. In any case it seems obvious that this interaction is not stable enough to keep them together as a complex in a SEC column. Then, they followed to see if the ATPase activity of RavA was affected by VirA. The results were explained as if different conformational states of RavA in the presence of ATP or ADP were affecting the interaction of RavA with VirA. This makes sense but, in physiological conditions, to which RavA conformation VirA binds?

Our BLI data demonstrates that ViaA binds to both apo-RavA, RavA + ADP and RavA + ATP. However, the affinity to RavA is increased by an order of magnitude after addition of ADP/ATP. The addition of ATP to RavA results in a ViaA binding curve that can be explained by the presence of two different states of RavA with different ViaA affinities. This is most probably the result of ongoing ATP hydrolysis by RavA during the binding experiment. It is important to note that RavA is present as a functional hexamer after incubation with ADP or ATP (Jessop *et al.*, 2020), and that it has the propensity to partially fall apart into monomers in the absence of ADP/ATP, thus providing a possible explanation for the notably lower affinity of ViaA for apo-RavA.

We propose to add following lines to the relevant paragraph regarding the strength and kinetics of the RavA-ViaA interaction in the main text:

“ViaA most likely solely interacts with hexameric RavA, in accordance with other characterized AAA+ ATPase and VWA domain-containing protein pairs (Sosnowski *et al.*, eLife 2018; Chen *et al.*, Cell 2018; Tsai *et al.*, PNAS 2020). Since RavA is present as a functional hexamer after incubation with ADP or ATP (Snider *et al.*, JBC, 2006) but has the propensity to partially fall apart into monomers in the absence of ADP/ATP (Jessop *et al.*, Communications Biology, 2020), this provides a possible explanation for the notably lower affinity of ViaA for apo-RavA.”

Regarding the remarks about the localization of these proteins as observed by microscopy, the explanations of the authors seem convincing. In general, the authors have answered to most of the concerns, adding new data (Supplementary Table I and Supplementary Figure 8) and splitting the previous Supp Table I in two (Supp. Tables 2 and 3). They also add more information in the Methods Section.

We hope that the additional clarifications provided here are satisfactory and the manuscript can now be accepted for publication.

Reviewer #2 (Remarks to the Author):

The authors convincingly replied to my concerns, I have no further comments or request.